# Label-efficient Segmentation via Affinity Propagation

**Wentong Li**[1*], **Yuqian Yuan**[1*], **Song Wang**[1], **Wenyu Liu**[1],
**Dongqi Tang**[2], **Jian Liu**[2], **Jianke Zhu**[1†], **Lei Zhang**[3]
[1]Zhejiang University    [2]Ant Group    [3]The Hong Kong Polytechnical University
https://LiWentomng.github.io/apro/

## Abstract

Weakly-supervised segmentation with label-efficient sparse annotations has attracted increasing research attention to reduce the cost of laborious pixel-wise labeling process, while the pairwise affinity modeling techniques play an essential role in this task. Most of the existing approaches focus on using the local appearance kernel to model the neighboring pairwise potentials. However, such a local operation fails to capture the long-range dependencies and ignores the topology of objects. In this work, we formulate the affinity modeling as an affinity propagation process, and propose a local and a global pairwise affinity terms to generate accurate soft pseudo labels. An efficient algorithm is also developed to reduce significantly the computational cost. The proposed approach can be conveniently plugged into existing segmentation networks. Experiments on three typical label-efficient segmentation tasks, *i.e.* box-supervised instance segmentation, point/scribble-supervised semantic segmentation and CLIP-guided semantic segmentation, demonstrate the superior performance of the proposed approach.

## 1   Introduction

Segmentation is a widely studied problem in computer vision, aiming at generating a mask prediction for a given image, *e.g.,* grouping each pixel to an object instance (*instance segmentation*) or assigning each pixel a category label (*semantic segmentation*). While having achieved promising performance, most of the existing approaches are trained in a fully supervised manner, which heavily depend on the pixel-wise mask annotations, incurring tedious labeling costs [1]. Weakly-supervised methods have been proposed to reduce the dependency on dense pixel-wise labels with label-efficient sparse annotations, such as points [2–4], scribbles [5–7], bounding boxes [8–11] and image-level labels [12–15]. Such methods make dense segmentation more accessible with less annotation costs for new categories or scene types.

Most of the existing weakly-supervised segmentation methods [16, 13, 8, 17, 3, 10] adopt the local appearance kernel to model the neighboring pairwise affinities, where spatially nearby pixels with similar color (*i.g.*, LAB color space [8, 3] or RGB color space [13, 16, 17, 10]) are likely to be in the same class. Though having proved to be effective, these methods suffer from two main limitations. First, the local operation cannot capture global context cues and capture long-range affinity dependencies. Second, the appearance kernel fails to take the intrinsic topology of objects into account, and lacks capability of detail preservation.

To address the first issue, one can directly enlarge the kernel size to obtain a large receptive filed. However, this will make the segmentation model insensitive to local details and increase the computational cost greatly. Some methods [14, 12] model the long-range affinity via random walk [18], but they cannot model the fine-grained semantic affinities. As for the second issue, the tree-based

---

*Equal contribution

†Correspondence author

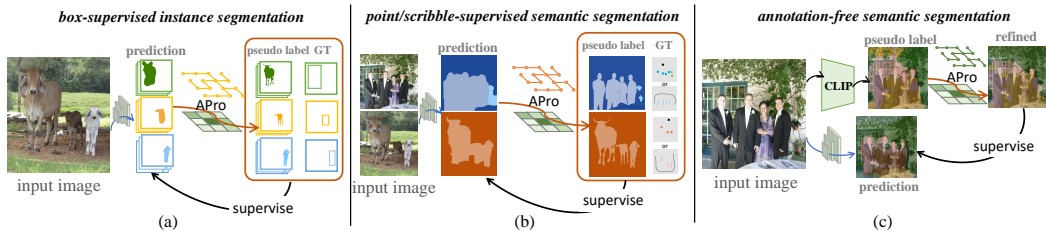

Figure 1: The proposed approach upon the typical weakly-supervised segmentation tasks with label-efficient annotations, including (a) box-supervised instance segmentation, (b) point/scribble-supervised semantic segmentation and (c) annotation-free semantic segmentation with CLIP pre-trained model.

approaches [7, 19] are able to preserve the geometric structures of objects, and employ the minimum spanning tree [20] to capture the pairwise relationship. However, the affinity interactions with distant nodes will decay rapidly as the distance increases along the spanning tree, which still focuses on the local nearby regions. LTF-V2 [21] enables the long-range tree-based interactions but it fails to model the valid pairwise affinities for label-efficient segmentation task.

With the above considerations, we propose a novel component, named Affinity Propagation (APro), which can be easily embedded in existing methods for label-efficient segmentation. Firstly, we define the weakly-supervised segmentation from a new perspective, and formulate it as a uniform affinity propagation process. The modelled pairwise term propagates the unary term to other nearby and distant pixels and updates the soft pseudo labels progressively. Then, we introduce the global affinity propagation, which leverages the topology-aware tree-based graph and relaxes the geometric constraints of spanning tree to capture the long-range pairwise affinity. With the efficient design, the $\mathcal{O}(N^2)$ complexity of brute force implementation is reduced to $\mathcal{O}(N \log N)$, and the global propagation approach can be performed with much less resource consumption for practical applications. Although the long-range pairwise affinity is captured, it inevitably brings in noises based on numerous pixels in a global view. To this end, we introduce a local affinity propagation to encourage the piece-wise smoothness with spatial consistency. The formulated APro can be embedded into the existing segmentation networks to generate accurate soft pseudo labels online for unlabeled regions. As shown in Fig. 1, it can be seamlessly plugged into the existing segmentation networks for various tasks to achieve weakly-supervised segmentation with label-efficient sparse annotations.

We perform experiments on three typical label-efficient segmentation tasks, *i.e.* box-supervised instance segmentation, point/scribble-supervised semantic segmentation and annotation-free semantic segmentation with pretrained CLIP model, and the results demonstrated the superior performance of our proposed universal label-efficient approach.

## 2 Related Work

**Label-efficient Segmentation.** Label-efficient segmentation, which is based on the weak supervision from partial or sparse labels, has been widely explored [1]. Different from semi-supervised settings [22, 23], this paper mainly focuses on the segmentation with sparse labels. In earlier literature [24–28], it primarily pertained to image-level labels. Recently, diverse sparse annotations have been employed, including *point*, *scribble*, *bounding box*, *image-level label* and the combinations of them. We briefly review the weakly-supervised instance segmentation, semantic segmentation and panoptic segmentation tasks in the following.

For weakly-supervised instance segmentation, box-supervised methods [16, 8, 17, 9, 29, 30, 11, 10, 31] are dominant and perform on par with fully-supervised segmentation approaches. Besides, the "points + bounding box" annotation can also achieve competitive performance [32, 33]. As for weakly-supervised semantic segmentation, previous works mainly focus on the point-level supervision [2, 34, 6] and scribble-level supervision [5, 35, 36], which utilize the spatial and color information of the input image and are trained with two stages. Liang *et al.* [7] introduced an effective tree energy loss based on the low-level and high-level features for point/scribble/block-supervised semantic segmentation. For semantic segmentation, the supervision of image-level labels has been well explored [12, 37, 15, 14]. Recently, some works have been proposed to make use of the large-scale pretrained CLIP model [38] to achieve weakly-supervised or annotation-free semantic

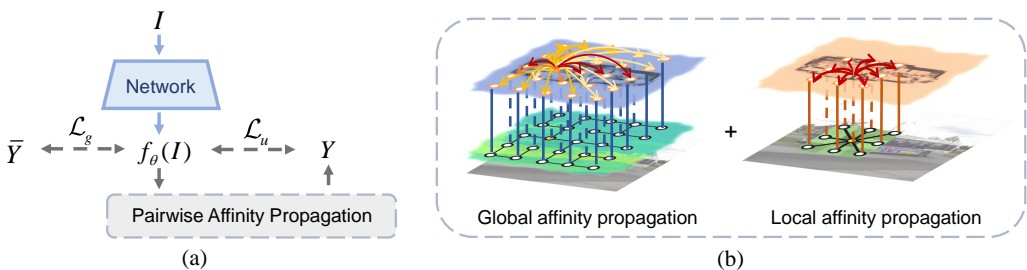

Figure 2: Overview of our `APro` approach. (a) The general weakly supervised segmentation framework with the proposed pairwise affinity propagation. (b) The proposed approach consists of global affinity propagation (GP) and local affinity propagation (LP) to generate accurate pseudo labels.

segmentation [39, 40]. In addition, weakly-supervised panoptic segmentation methods [3, 41, 4] with a single and multiple points annotation have been proposed. In this paper, we aim to develop a universal component for various segmentation tasks, which can be easily plugged into the existing segmentation networks.

**Pairwise Affinity Modeling.** Pairwise affinity modeling plays an important role in many computer vision tasks. Classical methods, like CRF [42], make full use of the color and spatial information to model the pairwise relations in the semantic labeling space. Some works use it as a post-processing module [43], while others integrate it as a jointly-trained part into the deep neural network [44, 45]. Inspired by CRF, some recent approaches explore the local appearance kernel to tap the neighboring pairwise affinities, where spatially nearby pixels with similar colors are more likely to fall into the same class. Tian *et al.* [8] proposed the local pairwise loss, which models the neighboring relationship in LAB color space. Some works [10, 16, 17] focus on the pixel relation in RGB color space directly, and achieve competitive performance. However, the local operation fails to capture global context cues and lacks the long-range affinity dependencies. Furthermore, the appearance kernel cannot reflect the topology of objects, missing the details of semantic objects. To model the structural pair-wise relationship, tree-based approaches [7, 19] leverage the minimum spanning tree [20] to capture the topology of objects in the image. However, the affinity interactions with distant nodes decay rapidly as the distance increases along the spanning tree. Besides, Zhang *et al.* [35] proposed an affinity network to convert an image to a weighted graph, and model the node affinities by the graph neural network (GNN). Different from these methods, in this work we model the pairwise affinity from a new affinity propagation perspective both globally and locally.

## 3 Methodology

In this section, we introduce our proposed affinity propagation (`APro`) approach to label-efficient segmentation. First, we define the problem and model the `APro` framework in Section 3.1. Then, we describe the detailed pairwise affinity propagation method in Section 3.2. Finally, we provide an efficient implementation of our method in Section 3.3.

### 3.1 Problem Definition and Modeling

Given an input image $I = \{x_i\}^N$ with $N$ pixels and its available sparse ground truth labeling $\bar{Y}$ (*i.e.* points, scribble, or bounding box), and let $f_\theta(I)$ be the output of a segmentation network with the learnable parameters $\theta$, the whole segmentation network can be regarded as a neural network optimization problem as follows:

$$\min_\theta \left\{ \mathcal{L}_g(f_\theta(I), \bar{Y}) + \mathcal{L}_u(f_\theta(I), Y) \right\}, \tag{1}$$

where $\mathcal{L}_g$ is a ground truth loss on the set of labeled pixels $\Omega_g$ with $\bar{Y}$ and $\mathcal{L}_u$ is a loss on the unlabeled regions $\Omega_u$ with the pseudo label $Y$. As shown in Fig. 2-(a), our goal is to obtain the accurate pseudo label $Y$ by leveraging the image pixel affinities for unlabeled regions.

As in the classic MRF/CRF model [46, 42], the unary term reflects the per-pixel confidence of assigning labels, while pairwise term captures the inter-pixel constraints. We define the generation of

pseudo label $Y$ as an affinity propagation process, which can be formulated as follows:

$$y_i = \frac{1}{z_i} \sum_{j \in \tau} \phi(x_j)\psi(x_i, x_j), \tag{2}$$

where $\phi(x_j)$ denotes the unary potential term, which is used to align $y_j$ and the corresponding network prediction based on the available sparse labels. $\psi(x_i, x_j)$ indicates the pairwise potential, which models the inter-pixel relationships to constrain the predictions and produce accurate pseudo labels. $\tau$ is the region with different receptive fields. $z_i$ is the summation of pairwise affinity $\psi(x_i, x_j)$ along with $j$ to normalize the response.

Notably, we unify both global and local pairwise potentials in an affinity propagation process formulated in Eq. 2. As shown in Fig. 2-(b), the global affinity propagation (GP) can capture the pairwise affinities with topological consistency in a global view, while the local affinity propagation (LP) can obtain the pairwise affinities with local spatial consistency. Through the proposed component, the soft pseudo labels $Y$ can be obtained. We assign each $y_i$ from GP and LP with the network prediction $p_i$ and directly employ the distance measurement function as the objective for unlabeled regions $\Omega_u$. Simple $L_1$ distance is empirically adopted in our implementation.

## 3.2 Pairwise Affinity Propagation

### 3.2.1 Global Affinity Propagation

We firstly provide a solution to model the global affinity efficiently based on the input image. Specifically, we represent an input image as a 4-connected planar graph $\mathcal{G}$, where each node is adjacent to up to 4 neighbors. The weight of the edge measures the image pixel distance between adjacent nodes. Inspired by tree-based approaches [47, 19], we employ the minimum spanning tree (MST) algorithm [20] to remove the edge with a large distance to obtain the tree-based sparse graph $\mathcal{G}_T$, *i.e.*, $\mathcal{G}_T \leftarrow \text{MST}(\mathcal{G})$ and $\mathcal{G}_T = \{\mathcal{V}, \mathcal{E}\}$, where $\mathcal{V} = \{\mathcal{V}_i\}^N$ is the set of nodes and $\mathcal{E} = \{\mathcal{E}_i\}^{N-1}$ denotes the set of edges.

Then, we model the global pairwise potential by iterating over each node. To be specific, we take the current node as the root of the spanning tree $\mathcal{G}_T$ and propagate the long-range affinities to other nodes. While the distant nodes along the spanning tree need to pass through nearby nodes along the path of spanning tree, the distance-insensitive `max` affinity function can alleviate this geometric constraint and relax the affinity decay for long-range nodes. Hence, we define the global pairwise potential $\psi_g$ as follows:

$$\psi_g(x_i, x_j) = \mathcal{T}(I_i, I_j) = \exp(-\max_{\forall (k,l) \in \mathbb{E}_{i,j}} \frac{w_{k,l}}{\zeta_g{}^2}), \tag{3}$$
$$\forall j \in \mathcal{V}$$

where $\mathcal{T}$ denotes the global tree. $\mathbb{E}_{i,j}$ is the set of edges in the path of $\mathcal{T}$ from node $j$ to node $i$. $w_{k,l}$ indicates the edge weight between the adjacent nodes $k$ and $l$, which is represented as the Euclidean distance between pixel values of two adjacent nodes, *i.e.*, $w_{k,l} = |I_k - I_l|^2$. $\zeta_g$ controls the degree of similarity with the long-range pixels. In this way, the global affinity propagation (GP) process to obtain the soft label $y^g$ can be formulated as follows:

$$y_i^g = \frac{1}{z_i^g} \sum_{j \in \mathcal{V}} \phi(x_j)\psi_g(x_i, x_j), \quad z_i^g = \sum_{j \in \mathcal{V}} \psi_g(x_i, x_j), \tag{4}$$

where interactions with distant nodes are performed over tree-based topology. In addition to utilizing low-level image, we empirically employ high-level feature as input to propagate semantic affinity.

### 3.2.2 Local Affinity Propagation

The long-range pairwise affinity is inevitably noisy since it is computed based on the susceptible image features in a global view. The spatially nearby pixels are more likely to have the same label, while they have certain difference in color and intensity, *etc*. Hence, we further introduce the local affinity propagation (LP) to promote the piece-wise smoothness. The Gaussian kernel is widely used to capture the local relationship among the neighbouring pixels in previous works [45, 8, 10]. Different from these works, we define the local pairwise affinity via the formulated affinity propagation process. The local pairwise term $\psi_s$ is defined as:

$$\psi_s(x_i, x_j) = \mathcal{K}(I_i, I_j) = \exp\left(\frac{-|I_i - I_j|^2}{\zeta_s^2}\right), \tag{5}$$
$$j \in \mathcal{N}(i)$$

where $\mathcal{K}$ denotes the Gaussian kernel, $\mathcal{N}(i)$ is the set containing all local neighbor pixels. The degree of similarity is controlled by parameter $\zeta_s$. Then the pseudo label $y^s$ can be obtained via the following affinity propagation:

$$y_i^s = \frac{1}{z_i^s} \sum_{j \in \mathcal{N}(i)} \phi(x_j)\psi_s(x_i, x_j), \;\; z_i^s = \sum_{j \in \mathcal{N}(i)} \psi_s(x_i, x_j), \tag{6}$$

where the local spatial consistency is maintained based on high-contrast neighbors. To obtain a robust segmentation performance, multiple iterations are required. Notably, our LP process ensures a fast convergence, which is $5\times$ faster than MeanField-based method [10, 42]. The details can be found in the experimental section 4.4.

### 3.3 Efficient Implementation

Given a tree-based graph $\mathcal{G}_T = \{\mathcal{V}, \mathcal{E}\}$ in the GP process, we define the maximum $\boldsymbol{w}$ value of the path through any two vertices as the transmission cost $\mathcal{C}$. One straightforward approach to get $y_i^g$ of vertex $i$ is to traverse each vertex $j$ by Depth First Search or Breadth First Search to get the transmission cost $\mathcal{C}_{i,j}$ accordingly. Consequently, the computational complexity required to acquire the entire output $y^g$ is $\mathcal{O}(N^2)$, making it prohibitive in real-world applications.

Instead of calculating and updating the transmission cost of any two vertices, we design a lazy update algorithm to accelerate the GP process. Initially, each node is treated as a distinct union, represented by $\mathcal{U}$. Unions are subsequently connected based on each edge $w_{k,l}$ in ascending order of $\boldsymbol{w}$. We show that when connecting two unions $\mathcal{U}_k$ and $\mathcal{U}_l$, $w_{k,l}$ is equivalent to the transmission cost for all nodes within $\mathcal{U}_k$ and $\mathcal{U}_l$. This is proved in the **supplementary material**.

To efficiently update values, we introduce a ***Lazy Propagation*** scheme. We only update the value of the root node and postpone the update of its descendants. The update information is retained in a *lazy tag* $\mathcal{Z}$ and is updated as follows:

$$\mathcal{Z}(\delta)_{k^*} = \mathcal{Z}(\delta)_{k^*} + \begin{cases} \exp(-w_{k,l}/\zeta_g{}^2)S(\delta)_l & \mathcal{U}_k.\text{rank} > \mathcal{U}_l.\text{rank}, \\ \exp(-w_{k,l}/\zeta_g{}^2)S(\delta)_l - \mathcal{Z}(\delta)_{l^*} & \text{otherwise}, \end{cases} \tag{7}$$

where $S(\delta)_i = \sum_{j \in \mathcal{U}_i} \delta_j$, $\delta$ means different inputs, including the dense prediction $\phi(x)$ and all-one matrix $\Lambda$. $k^*/l^*$ denotes the root node of node $k/l$.

Once all unions are connected, the lazy tags can be propagated downward from the root node to its descendants. For the descendants, the global affinity propagation term is presented as follows:

$$LProp(\delta)_i = \delta_i + \sum_{r \in Asc_{\mathcal{G}_T}(i) \cup \{i\}} \mathcal{Z}(\delta)_r, \tag{8}$$

where $Asc_{\mathcal{G}_T}(i)$ represents the ascendants of node $i$ in the tree $\mathcal{G}_T$. As shown in Algorithm 1, the disjoint-set data structure is employed to implement the proposed algorithm. In our implementation, a Path Compression strategy is applied, connecting each node on the path directly to the root node. Consequently, it is sufficient to consider the node itself and its parent node to obtain $LProp$.

**Time complexity.** For each channel of the input, the average time complexity of sorting is $\mathcal{O}(N \log N)$. In the merge step, we utilize the Path Compression and Union-by-Rank strategies, which have a complexity of $\mathcal{O}(\alpha(N))$[48]. After merging all the concatenated blocks, the lazy tags can be propagated in $\mathcal{O}(N)$ time. Hence, the overall complexity is $\mathcal{O}(N \log N)$. Note that the batches and channels are independent of each other. Thus, the algorithm can be executed in parallel for both batches and channels for practical implementations. As a result, the proposed algorithm reduces the computational complexity dramatically.

## 4 Experiments

### 4.1 Weakly-supervised Instance Segmentation

**Datasets.** As in prior arts [8, 9, 16, 17], we conduct experiments on two widely used datasets for the *weakly box-supervised instance segmentation* task:
- COCO [49], which has 80 classes with 115K `train2017` images and 5K `val2017` images.

**Algorithm 1:** Algorithm for GP process

---

**Input:** Tree $\mathcal{G}_T \in \mathbb{N}^{e \times 2}$; Pairwise distance $\boldsymbol{w} \in \mathbb{R}^N$; Dense predictions $\phi(x) \in \mathbb{R}^N$;
      Vertex num $N$; Edge num $e = N - 1$; Set of vertices $\mathcal{V}$.

**Output:** $y^g \in \mathbb{R}^N$.

$\Lambda \leftarrow \mathbf{1} \in \mathbb{R}^N$

$F \leftarrow \{0, 1, 2, ..., N - 1\}$             ▷ Initialize each vertex as a connected block

Sort $\{\mathcal{G}_T, \boldsymbol{w}\}$ in ascending order of $\boldsymbol{w}$.             ▷ Quick Sort

**for** $(k, l) \in \mathcal{G}_T, w_i \in \boldsymbol{w}$ **do**

    $a \leftarrow \text{find}(k), b \leftarrow \text{find}(l)$             ▷ Find the root node with Path Compression

    Update$\{\mathcal{Z}(\phi)_a, \mathcal{Z}(\Lambda)_a, \mathcal{Z}(\phi)_b, \mathcal{Z}(\Lambda)_b\}$             ▷ Add lazy tag

    **if** $S_a < S_b$ **then**

        swap$(a, b)$             ▷ Merge by Rank

    $F_b \leftarrow a$             ▷ Merge two connected blocks

**for** $v \in \mathcal{V}$ **do**

    $p \leftarrow \text{find}(v)$

    **for** $\delta \in \{\phi, \Lambda\}$ **do**

        **if** $p = v$ **then**

           $LProp(\delta)_v = \mathcal{Z}(\delta)_v + \delta_v$

        **else**

           $LProp(\delta)_v = \mathcal{Z}(\delta)_p + \mathcal{Z}(\delta)_v + \delta_v$

    $y^g_v = \frac{LProp(\phi)_v}{LProp(\Lambda)_v}$             ▷ Normalization

**return** $y^g$

---

- Pascal VOC [43] augmented by SBD [50] based on the original Pascal VOC 2012 [51], which has 20 classes with 10,582 `trainaug` images and 1,449 `val` images.

**Base Architectures and Competing Methods.** In the evaluation, we apply our proposed `APro` to two representative instance segmentation architectures, SOLOv2 [52] and Mask2Former [53], with different backbones (*i.e.*, ResNet [54], Swin-Transformer [55]) following Box2Mask [11]. We compare our approach with its counterparts that model the pairwise affinity based on the image pixels without modifying the base segmentation network for box-supervised setting. Specifically, the compared methods include Pairwise Loss [8], TreeEnergy Loss [7] and CRF Loss [10]. For fairness, we re-implement these models using the default setting in MMDetection [56].

**Implementation Details.** We follow the commonly used training settings on each dataset as in MMDetection [56]. All models are initialized with ImageNet [57] pretrained backbone. For SOLOv2 framework [52], the scale jitter is used, where the shorter image side is randomly sampled from 640 to 800 pixels. For Mask2Former framework [53], the large-scale jittering augmentation scheme [58] is employed with a random scale sampled within range [0.1, 2.0], followed by a fixed size crop to 1024×1024. The initial learning rate is set to $10^{-4}$ and the weight decay is 0.05 with 16 images per mini-batch. The box projection loss [8, 9] is employed to constrain the network prediction within the bounding box label as the unary term $\phi$. COCO-style mask AP (%) is adopted for evaluation.

**Quantitative Results.** Table 1 shows the quantitative results. We compare the approaches with the same architecture for fair comparison. The state-of-the-art methods are listed for reference. One can see that our `APro` method outperforms its counterparts across Pascal VOC and COCO datasets.

- Pascal VOC [43] `val`. Under the SOLOv2 framework, our approach achieves 37.1% AP and 38.4% AP with 12 epochs and 36 epochs, respectively, outperforming other methods by 1.4%-2.5% mask AP with ResNet-50. With the Mask2Former framework, our approach also outperforms its counterparts. Furthermore, with the Swin-L backbone [55], our proposed approach achieves very promising performance, 49.6% mask AP with 50 epochs.

- COCO [49] `val`. Under the SOLOv2 framework, our approach achieves 32.0% AP and 32.9% AP with 12 epochs and 36 epochs, and surpasses its best counterpart by 1.0% AP and 0.4% AP using ResNet-50, respectively. Under the Mask2Former framework, our method still achieves the best performance with ResNet-50 backbone. Furthermore, equipped with stronger backbones, our approach obtains more robust performance, achieving 38.0% mask AP with ResNet-101, and 41.0% mask AP using Swin-L backbone.

Table 1: Quantitative results (§4.1) on Pascal VOC [43] and COCO `val` [49] with mask AP(%).

| Method | Backbone | #Epoch | Pascal VOC | | | COCO | | |
|---|---|---|---|---|---|---|---|---|
| | | | AP | $AP_{50}$ | $AP_{75}$ | AP | $AP_{50}$ | $AP_{75}$ |
| BBTP [NeurIPS19] [16] | ResNet-101 | 12 | 23.1 | 54.1 | 17.1 | 21.1 | 45.5 | 17.2 |
| BoxInst [CVPR21] [8] | ResNet-50 | 36 | 34.3 | 58.6 | 34.6 | 31.8 | 54.4 | 32.5 |
| DiscoBox [ICCV21] [17] | ResNet-50 | 36 | - | 59.8 | 35.5 | 31.4 | 52.6 | 32.2 |
| BoxLevelset [ECCV22] [9] | ResNet-50 | 36 | 36.3 | 64.2 | 35.9 | 31.4 | 53.7 | 31.8 |
| *SOLOv2 Framework* | | | | | | | | |
| Pairwise Loss [CVPR21] [8] | ResNet-50 | 12 | 35.7 | 64.3 | 35.1 | 31.0 | 52.8 | 31.5 |
| TreeEnergy Loss [CVPR22] [7] | ResNet-50 | 12 | 35.0 | 64.4 | 34.7 | 30.9 | 52.9 | 31.3 |
| CRF Loss [CVPR23] [10] | ResNet-50 | 12 | 35.0 | 64.7 | 34.9 | 30.9 | 53.1 | 31.4 |
| APro(Ours) | ResNet-50 | 12 | **37.1** | **65.1** | **37.0** | **32.0** | **53.4** | **32.9** |
| Pairwise Loss [CVPR21] [8] | ResNet-50 | 36 | 36.5 | 63.4 | 38.1 | 32.4 | 54.5 | 33.4 |
| TreeEnergy Loss [CVPR22] [7] | ResNet-50 | 36 | 36.1 | 63.5 | 36.1 | 31.4 | 54.0 | 31.2 |
| CRF Loss [CVPR23] [10] | ResNet-50 | 36 | 35.9 | 64.0 | 35.7 | 32.5 | 54.9 | 33.2 |
| APro(Ours) | ResNet-50 | 36 | 38.4 | 65.4 | 39.8 | 32.9 | 55.2 | 33.6 |
| APro(Ours) | ResNet-101 | 36 | **40.5** | **67.9** | **42.6** | **34.3** | **57.0** | **35.3** |
| *Mask2Former Framework* | | | | | | | | |
| Pairwise Loss [CVPR21] [8] | ResNet-50 | 12 | 35.2 | 62.9 | 33.9 | 33.8 | 57.1 | 34.0 |
| TreeEnergy Loss [CVPR22] [7] | ResNet-50 | 12 | 36.0 | 65.0 | 34.3 | 33.5 | 56.7 | 33.7 |
| CRF Loss [CVPR23] [10] | ResNet-50 | 12 | 35.7 | 64.3 | 35.2 | 33.5 | 57.5 | 33.8 |
| APro(Ours) | ResNet-50 | 12 | 37.0 | 65.1 | 37.0 | 34.4 | 57.7 | 35.3 |
| APro(Ours) | ResNet-50 | 50 | 42.3 | 70.6 | 44.5 | 36.1 | 62.0 | 36.7 |
| APro(Ours) | ResNet-101 | 50 | 43.6 | 72.0 | 45.7 | 38.0 | 63.6 | 38.7 |
| APro(Ours) | Swin-L | 50 | **49.6** | **77.6** | **53.1** | **41.0** | **68.3** | **41.9** |

**Qualitative Results.** Fig. 3 illustrates the visual comparisons on affinity maps of our APro and other approaches, and Fig. 4 compares the segmentation results. One can clearly see that our method captures accurate pairwise affinity with object's topology and yields more fine-grained predictions.

## 4.2 Weakly-supervised Semantic Segmentation

**Datasets.** We conduct experiments on the widely-used Pascal VOC2012 dataset [51], which contains 20 object categories and a background class. As in [6, 7], the augmented Pascal VOC dataset is adopted here. The `point` [2] and `scribble` [5] annotations are employed for *weakly point-supervised and scribble-supervised settings*, respectively.

**Implementation Details.** As in [7], we adopt LTF [19] as the base segmentation model. The input size is $512 \times 512$. The SGD optimizer with momentum of 0.9 and weight decay of $10^{-4}$ is used. The initial learning rate is 0.001, and there are 80k training iterations. The same data augmentations as in [7] are utilized. We employ the partial cross-entropy loss to make full use of the available point/scribble labels and constrain the unary term. ResNet-101 [54] pretrained on ImageNet [57] is adopted as backbone network for all methods.

Table 2: Quantitative results (§4.2) on Pascal VOC2012 [51] `val` with mean IoU(%).

| Method | Backbone | Supervision | CRF Post. | **mIoU** |
|---|---|---|---|---|
| †KernelCut Loss [ECCV18] [6] | DeepLabV2 | | ✓ | 57.0 |
| *TEL [CVPR22] [7] | LTF | Point | ✗ | 66.8 |
| APro(Ours) | LTF | | ✗ | **67.7** |
| †NormCut Loss [CVPR18] [59] | DeepLabV2 | | ✓ | 74.5 |
| †DenseCRF Loss [ECCV18] [6] | DeepLabV2 | | ✓ | 75.0 |
| †KernelCut Loss [ECCV18] [6] | DeepLabV2 | Scribble | ✓ | 75.0 |
| †GridCRF Loss [ICCV19] [36] | DeepLabV2 | | ✗ | 72.8 |
| PSI [ICCV21] [36] | DeepLabV3 | | ✗ | 74.9 |
| *TEL [CVPR22] [7] | LTF | | ✗ | 76.2 |
| APro(Ours) | LTF | | ✗ | **76.6** |

†:adopting multi-stage training, *:our re-implementation.

**Quantitative Results.** As shown in Table 2, we compare our APro approach with the state-of-the-art methods on point-supervised and scribble-supervised semantic segmentation, respectively.

- Point-wise supervision. With DeepLabV2 [43], KernelCut Loss [6] achieves 57.0% mIoU. Equipped with LTF [19], TEL [7] achieves 66.8% mIoU. Our APro achieves 67.7% mIoU, outperforming the previous best method TEL [7] by 0.9% mIoU.
- Scribble-wise supervision. The scribble-supervised approaches are popular in weakly supervised semantic segmentation. We apply the proposed approach under the single-stage training framework without calling for CRF post-processing during testing. Compared with the state-of-the-art methods, our approach achieves better performance with 76.6% mIoU.

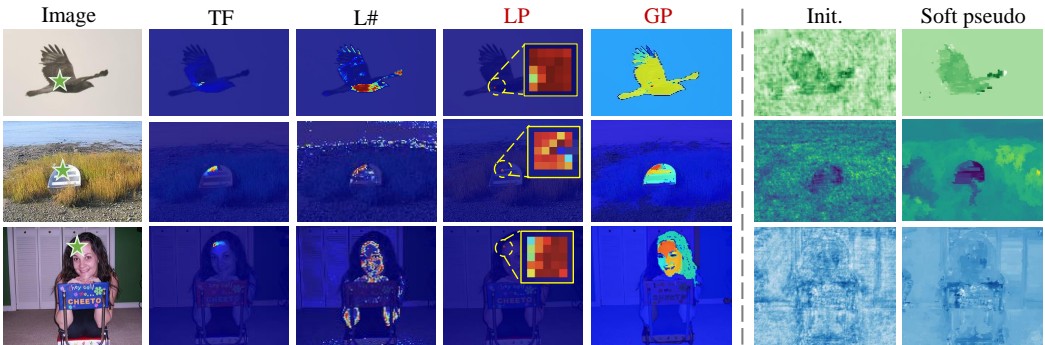

| Image | TF | L# | LP | GP | | Init. | Soft pseudo |

Figure 3: Visual comparisons of pairwise affinity maps based on the RGB image for a specific position (green star), including TreeFilter (TF) [19], local kernels with full image size (L#), and our presented LP and GP processes. The GP process can capture the long-range pairwise affinity with object's topology, while LP retrieves the local similarities. Our `APro` approach smooths the noisy initial network predictions (init.) to obtain cleaner soft pseudo labels.

## 4.3 CLIP-guided Semantic Segmentation

**Datasets.** To more comprehensively evaluate our proposed approach, we conduct experiments on *CLIP-guided annotation-free semantic segmentation* with three widely used datasets:

- Pascal VOC 2012 [51] introduced in Section 4.2.

- Pascal Context [60], which contains 59 foreground classes and a background class with 4,996 `train` images and 5,104 `val` images.

- COCO-Stuff [61], which has 171 common semantic object/stuff classes on 164K images, containing 118,287 `train` images and 5,000 `val` images.

**Base Architectures and Backbones.** We employ MaskCLIP+ [40] as our base architecture, which leverages the semantic priors of pretrained CLIP [38] model to achieve the annotation-free dense semantic segmentation. In the experiments, we couple MaskCLIP+ with our `APro` approach under ResNet-50, ResNet-50×16 and ViT-B/16 [62]. The dense semantic predictions of MaskCLIP [40] are used as the unary term, and our proposed method can refine it and generate more accurate pseudo labels for training target networks.

**Implementation Details.** For fair comparison, we keep the same settings as MaskCLIP+ [40]. We keep the text encoder of CLIP unchanged and take prompts with target classes as the input. For text embedding, we feed prompt engineered texts into the text encoder of CLIP with 85 prompt templates, and average the results with the same class. For ViT-B/16, the bicubic interpolation is adopted for the pretrained positional embeddings. The initial learning rate is set to $10^{-4}$. We train all models with batch size 32 and 2k/4k/8k iterations. DeepLabv2-ResNet101 is used as the backbone.

**Quantitative Results.** Table 3 compares our approach with MaskCLIP+ [40] for annotaion-free semantic segmentation. We have the following observations.

Table 3: Quantitative results (§4.3) on Pascal VOC2012 [51] `val`, Pascal Context [60] `val`, and COCO-Stuff [61] `val` with mean IoU (%).

| Method | CLIP Model | VOC2012 | Context | COCO. |
|---|---|---|---|---|
| MaskCLIP+ [ECCV22] [40] | ResNet-50 | 58.0 | 23.9 | 13.6 |
| APro(Ours) | | **61.6** ↑3.6 | **25.4** ↑1.5 | **14.6** ↑1.0 |
| MaskCLIP+ [ECCV22] [40] | ResNet-50×16 | 67.5 | 25.2 | 17.3 |
| APro(Ours) | | **70.4** ↑2.9 | **26.5** ↑1.3 | **18.2** ↑0.9 |
| MaskCLIP+ [ECCV22] [40] | ViT-B/16 | 73.6 | 31.1 | 18.0 |
| APro(Ours) | | **75.1** ↑1.5 | **32.6** ↑1.5 | **19.5** ↑1.5 |

- Pascal VOC2012 [51] `val`. With ResNet-50 as the image encoder in pretrained CLIP model, our approach outperforms MaskCLIP+ by 3.6% mIoU. With ResNet-50×16 and ViT-B/16 as the image encoders, our method surpasses MaskCLIP+ by 2.9% and 1.5% mIoU, respectively.

- Pascal Context [60] `val`. Our proposed method outperforms MaskCLIP+ consistently with different image encoders (about +1.5% mIoU).

- COCO-Stuff [61] `val`. COCO-Stuff consists of hundreds of semantic categories. Our method still brings +1.0%, +0.9% and +1.5% performance gains over MaskCLIP+ with ResNet-50, ResNet-50×16 and ViT-B/16 image encoders, respectively.

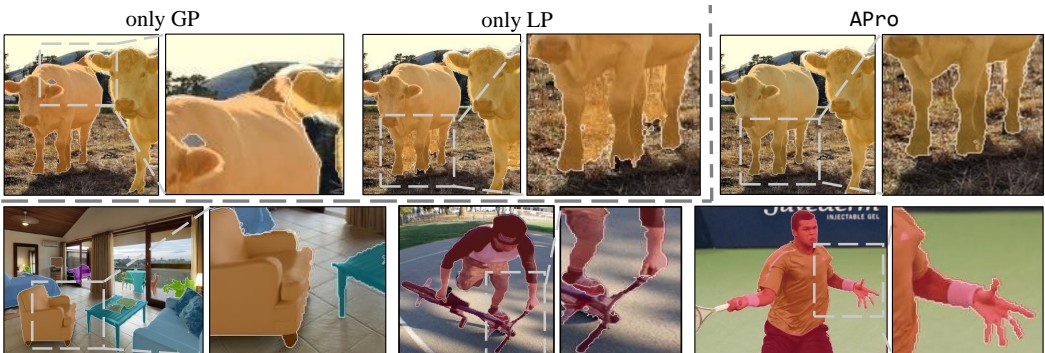

Figure 4: Qualitative results with different pairwise affinity terms on weakly supervised instance segmentation. Our method only with GP process preserves details without local consistency, while the model with only LP process encourages the local smoothness without topology-wise details. Our `APro` approach yields high-quality predictions with fine-grained details.

## 4.4 Diagnostic Experiments

For in-depth analysis, we conduct ablation studies on Pascal VOC [51] upon the weakly box-supervised instance segmentation task.

**Unary and Pairwise Terms.** Table 4 shows the evaluation results with different unary and pairwise terms. When using the unary term only, our method achieves 25.9% AP. When the global pairwise term is employed, our method achieves a much better performance

Table 4: Effects of unary and pairwise terms.

| Unary | Global Pairwise | Local Pairwise | AP | $AP_{50}$ | $AP_{75}$ |
|---|---|---|---|---|---|
| ✓ | | | 25.9 | 57.0 | 20.4 |
| ✓ | ✓ | | 36.3 | 63.9 | 37.0 |
| ✓ | | ✓ | 36.0 | 64.3 | 35.6 |
| ✓ | ✓ | ✓ | **38.4** | **65.4** | **39.8** |

of 36.3% AP. Using the local pairwise term only, our method obtains 36.0% AP. When both the global and local pairwise terms are adopted, our method achieves the best performance of 38.4% AP.

**Tree-based Long-range Affinity Modeling.** The previous works [19, 7] explore tree-based filters for pairwise relationship modeling. Table 5 compares our method with them. TreeFilter can capture the relationship with dis-

Table 5: Comparisons with tree-based methods.

| Method | AP | $AP_{50}$ | $AP_{75}$ |
|---|---|---|---|
| TreeFilter [19] | 36.1 | 63.5 | 36.1 |
| TreeFilter [19] + Local Pairwise | 36.8 | 64.4 | 36.5 |
| Global + Local Pairwise (`Ours`) | **38.4** | **65.4** | **39.8** |

tant nodes to a certain extent (see Fig. 3). Directly using TreeFilter as the pairwise term leads to 36.1% AP. By combining TreeFilter with our local pairwise term, the model obtains 36.8%AP. In comparison, our proposed approach achieves 38.4% AP.

**Iterated Local Affinity Modeling.** We evaluate our local affinity propagation (LP) with different iterations, and compare it with the classical MeanField method [10, 42]. Table 6 reports the comparison results. Our `APro` with the LP process achieves 36.0% AP after 20 iterations. However, replacing our local affinity propagation with MeanFiled-based method [10] costs 100 iterations to obtain 35.9% AP. This indicates that our LP method possesses the attribute of fast convergence.

Table 6: Comparisons on local pairwise affinity modeling.

| LP(`Ours`) | | MeanField[10] | |
|---|---|---|---|
| Iteration | AP | Iteration | AP |
| 10 | 35.8 | 20 | 35.2 |
| 20 | **36.0** | 30 | 35.5 |
| 30 | 35.7 | 50 | 35.5 |
| 50 | 35.6 | 100 | **35.9** |

**Soft Pseudo-label Generation.** With the formulated GP and LP methods, we study how to integrate them to generate the soft pseudo-labels in Table 7. We can cascade GP and LP sequentially to refine the pseudo labels. Putting GP before LP (denoted as GP-LP-C) achieves 36.8% AP, and putting LP before GP (denoted as LP-GP-C) performs better with 37.7% AP. In addition, we can use

Table 7: Generation of soft pseudo labels.

| Method | AP | $AP_{50}$ | $AP_{75}$ |
|---|---|---|---|
| GP-LP-C | 36.8 | 63.7 | 37.8 |
| LP-GP-C | 37.7 | 65.1 | 39.1 |
| GP-LP-P | **38.4** | **65.4** | **39.8** |

GP and LP in parallel (denoted as GP-LP-P) to produce two pseudo labels, and employ both of them to optimize the segmentation network with $L_1$ distance. Notably, GP-LP-P achieves the best performance with 38.4% mask AP. This indicates that our proposed affinity propagation in global and local views are complementary for optimizing the segmentation network.

**Runtime Analysis.** We report the average runtime of our method in Table 8. The experiment is conducted on a single GeForce RTX 3090 with batch size 1. Here we report the average runtime for one GP process duration of an epoch on the Pascal VOC dataset. When directly using Breadth First Search for each node with $N$ times, the runtime is $4.3 \times 10^3$ ms with $\mathcal{O}(N^2)$ time complexity. While employing the proposed efficient implementation, the runtime is only 0.8 ms with $\mathcal{O}(N \log N)$ time complexity. This demonstrates that the proposed efficient implementation reduces the computational complexity dramatically.

Table 8: Average runtime (ms) with and without the efficient implementation.

| Effic. Imple. | Ave. Runtime |
|---|---|
| ✗ | $4.3 \times 10^3$ |
| ✓ | 0.8 |

## 5 Conclusion

In this work, we proposed a novel universal component for weakly-supervised segmentation by formulating it as an affinity propagation process. A global and a local pairwise affinity term were introduced to generate the accurate soft pseudo labels. An efficient implementation with the light computational overhead was developed. The proposed approach, termed as APro, can be embedded into the existing segmentation networks for label-efficient segmentation. Experiments on three typical label-efficient segmentation tasks, *i.e.*, box-supervised instance segmentation, point/scribble-supervised semantic segmentation and CLIP-guided annotation-free semantic segmentation, proved the effectiveness of proposed method.

## Acknowledgments

This work is supported by National Natural Science Foundation of China under Grants (61831015). It is also supported by the Information Technology Center and State Key Lab of CAD&CG, Zhejiang University.

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
