# Label-efficient Segmentation via Affinity Propagation
## *Supplementary Material*

**Wentong Li**[1*]**, Yuqian Yuan**[1*]**, Song Wang**[1]**, Wenyu Liu**[1]**,**
**Dongqi Tang**[2]**, Jian Liu**[2]**, Jianke Zhu**[1†]**, Lei Zhang**[3]
[1]Zhejiang University    [2]Ant Group    [3] The Hong Kong Polytechnical University

In this document, we provide more details, additional experimental results and discussions on our approach. The supplementary material is organized as follows:

- §A: more details on the efficient implementation;
- §B: additional graphical illustration;
- §C: more performance comparisons;
- §D: additional visualization results;
- §E: discussions.

## A    More Details on the Efficient Implementation.

In this section, we first present the proofs of our claims about transmission cost and lazy propagation in our proposed lazy update algorithm. Then, we provide the pseudo-code of the *find* function in Algorithm 1 of the main paper. The symbols in this document follow the same definitions as the main paper.

### A.1    Proofs on Transmission Cost and Lazy Propagation

**Lemma 1.** *Given edge $\mathcal{E}_{(k,l)}$ in $\mathcal{G}_T$ with edge weight $w_{k,l}$, $\forall a \in \mathcal{U}_k$, $b \in \mathcal{U}_l$, the transmission cost between vertex a and b is $w_{k,l}$.*

*Proof.* Since there are no loops in the tree, the shortest path between any two vertices is unique. Therefore, there exists a path $a{-}k$ in $\mathcal{U}_k$ that connects vertices $a$ and $k$, and a path $b{-}l$ that connects $b$ and $l$ in $\mathcal{U}_l$. When connecting unions $\mathcal{U}_k$ and $\mathcal{U}_l$ through edge $\mathcal{E}_{(k,l)}$, there is exactly a single path connecting $a$ and $b$, denoted as $a{-}k{-}l{-}b$. As the weight $w$ is sorted in ascending order, for any edge $\mathcal{E}_i$ with $w_i$ between $a{-}k$ in $\mathcal{U}_k$, we have $w_i \leq w_{k,l}$. The same conclusion applies to $l{-}b$. Hence, the maximum weight in path $a{-}k{-}l{-}b$ is $w_{k,l}$. Consequently, once $k$ and $l$ are connected, $w_{k,l}$ is equivalent to the transmission cost for all nodes within $\mathcal{U}_k$ and $\mathcal{U}_l$.

$\square$

**Lemma 2.** *When connecting vertices $k$ and $l$, lazy tags $\mathcal{Z}(\delta)_{k*}$ and $\mathcal{Z}(\delta)_{l*}$ can be updated as follows:*

$$\mathcal{Z}(\delta)_{k*} = \mathcal{Z}(\delta)_{k*} + \begin{cases} exp(-w_{k,l}/{\zeta_g}^2)S(\delta)_l & \mathcal{U}_k.rank > \mathcal{U}_l.rank, \\ exp(-w_{k,l}/{\zeta_g}^2)S(\delta)_l - \mathcal{Z}(\delta)_{l*} & otherwise. \end{cases} \tag{1}$$

*Proof.* Given $a \in \mathcal{U}_k$, for $\forall b \in \mathcal{U}_l$, the transmission cost between $a$ and $b$ is $w_{k,l}$. We have:

---

*Equal contribution
†Correspondence author

37th Conference on Neural Information Processing Systems (NeurIPS 2023).

$$\Delta LProp(\delta)_a = \sum_{i \in \mathcal{U}_l} (\exp(-w_{k,l}/\zeta_g{}^2)\delta_i) = \exp(-w_{k,l}/\zeta_g{}^2)S(\delta)_l. \tag{2}$$

First, let $\mathcal{U}_k$.rank $> \mathcal{U}_l$.rank. When merging unions $\mathcal{U}_k$ and $\mathcal{U}_l$, we choose $k^*$ as the root node and let $l^*$ be its descendant. There is:

$$\Delta LProp(\delta)_a = \Delta\mathcal{Z}(\delta)_{k^*}, \tag{3}$$

$$\therefore \Delta\mathcal{Z}(\delta)_{k^*} = \exp(-w_{k,l}/\zeta_g{}^2)S(\delta)_l. \tag{4}$$

Second, let $\mathcal{U}_k$.rank $\leq \mathcal{U}_l$.rank. When merging unions $\mathcal{U}_k$ and $\mathcal{U}_l$, we instead choose $l^*$ as the root node and let $k^*$ be its descendant. Then we have:

$$\Delta LProp(\delta)_a = \mathcal{Z}(\delta)_{l^*} + \Delta\mathcal{Z}(\delta)_{k^*} = \exp(-w_{k,l}/\zeta_g{}^2)S(\delta)_l, \tag{5}$$

$$\therefore \Delta\mathcal{Z}(\delta)_{k^*} = \exp(-w_{k,l}/\zeta_g{}^2)S(\delta)_l - \mathcal{Z}(\delta)_{l^*}. \tag{6}$$

$\square$

## A.2 Pseudo Code

The pseudo-code of the *find* function is shown in Algorithm A1, which finds the root rode with Path Compression.

---

**Algorithm A1:** Pseudo-code of the *find* function with Path Compression

---

```
/*
fa: the parent of the i-th node, shape: (N)
tag: the lazy tag of numbers
ptag: the lazy tag of predictions
*/

int find(int x){
/*
x: the node index to query
return: the root node of x
*/
    int fx = fa[x];
    if(fx == x)
        return x;

    fa[x] = find(fx); // Path Compression
    if(fa[x] != fx){
        tag[x] += tag[fx]; // Downlink lazy tag
        ptag[x] += ptag[fx];
    }

    return fa[x];
}
```

---

# B Additional Graphical Illustration

To facilitate a better comprehension, we provide a detailed graphical illustration in Fig. A1 to describe our global affinity propagation process. Initially, an input image is represented as a 4-connected planar graph. Subsequently, the Minimum Spanning Tree (MST) is constructed based on the edge weights to obtain the tree-based graph $\mathcal{G}_T$. $\psi_g(x_i, x_j)$ is calculated as $exp(-d)$, where $d$ is the maximum value along the path $E_{i,j}$ from node $x_i$ to node $x_j$. This pairwise similarity $\psi_g(x_i, x_j)$ is then multiplied by the unary term to obtain soft pseudo predictions.

Note that Fig. A1 serves purely as a visual illustration of our method. In the implementation, it is unnecessary to compute as it explicitly. As detailed in Section 3.3 of main paper, we alternatively design a lazy propagation scheme to efficiently update these values.

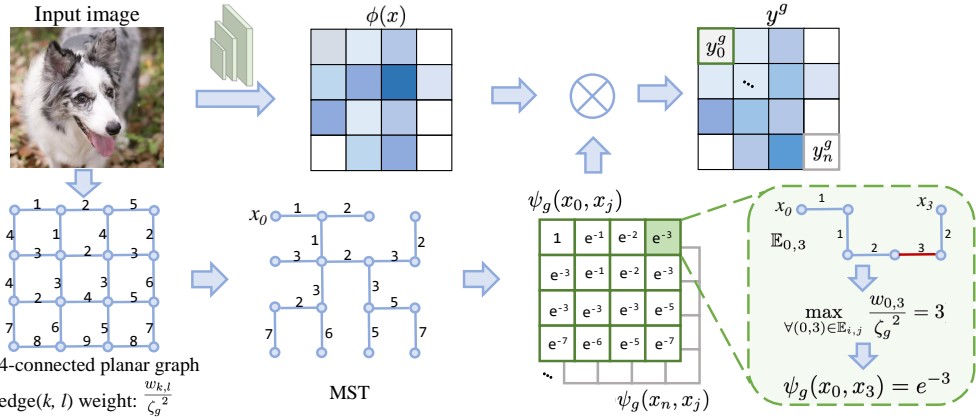

Figure A1: The graphical illustration of the detailed process of global affinity propagation. In the green dashed box, we present the calculation of $\psi_g(x_0, x_3)$ as a simple example.

## C  More Performance Comparisons

For annotation-free semantic segmentation with pretrained CLIP model, Key Smoothing (KS) proposed in MaskCLIP [1] also aims to realize the global affinity propagation. To better explore their efforts, we conduct detailed comparisons between KS and our `APro` method based on training-free MaskCLIP [1]. The experimental results are shown in Table A1.

Both KS and our `APro` method bring performance gains. Compared with KS, `APro` achieves better performance with different CLIP-based models. Especially, for ViT-B/16 model, our approach outperforms KS by +5.04% mIoU on Pascal Context and +2.90% mIoU on COCO, repectively. Equipped with Prompt Denoising (PD), the models could achieve further improvements.

We have the following further discussions: KS relies on the calculation of key feature similarities, which predominantly stems from high-level features of CLIP and computes pairwise terms within each pair of patches. Compared with KS of MaskCLIP, our method is built on a tree-based graph derived from low-level images, which is capable of capturing finer topological details.

Table A1: Quantitative results on Pascal Context [2] `val` and COCO-Stuff [3] `val` with mean IoU (%).

| Method | CLIP Model | Context | COCO. |
|---|---|---|---|
| MaskCLIP [1] | | 18.46 | 10.17 |
| +KS | ResNet-50 | 21.0 | 12.42 |
| +APro(Ours) | | **21.67** | **12.70** |
| MaskCLIP [1] | | 21.57 | 13.55 |
| +KS | ResNet-50×16 | 22.65 | 15.50 |
| +APro(Ours) | | **24.03** | **16.30** |
| MaskCLIP [1] | | 21.68 | 12.51 |
| +KS | | 23.87 | 13.79 |
| +KS+PD | ViT-B/16 | 25.45 | 14.62 |
| +APro(Ours) | | 28.91 | 16.69 |
| +APro(Ours) +PD | | **29.42** | **16.71** |

## D  Additional Visualization Results

To further show the performance of our proposed `APro` approach, we provide more visualization results. Fig. A2 shows the qualitative comparisons with the state-of-the-art methods upon box-supervised instance segmentation task [4, 5, 6]. It can be seen that our proposed `APro` approach is able to generate more accurate boundaries. For weakly-supervised semantic segmentation, we compare our method with the prior art TEL [7] upon point-wise supervision in Fig. A3. `APro` captures the fine-grained details of objects with the fitting boundaries. As for CLIP-guided annotation-free semantic segmentation, Fig. A4 provides the comparison results with MaskCLIP+ [1]. It can be observed that our approach eliminates the noisy predictions from the pretrained CLIP model effectively, achieving high-quality mask predictions. In addition, Fig. A5 provides the qualitative results of our method on general COCO dataset.

# E    Discussions

**Asset License and Consent.**    We use four image segmentation datasets, *i.e.*, COCO [8], Pascal VOC 2012 [9], COCO-Stuff [3] and Pascal Context [2], which are all publicly and freely available for academic research.    We implement all models with MMDetection [10], MMSegmentation [11] and openseg.pytorch [12] codebases.    COCO (https://cocodataset.org/) is released under the CC BY 4.0.    Pascal VOC 2012 (http://host.robots.ox.ac.uk/pascal/VOC/voc2012/) is released under the Flickr Terms of use for images.    COCO-Stuff v1.1 (https://github.com/nightrome/cocostuff) is released under the Flickr Terms of use for images and the CC BY 4.0 for annotations.    MMDetection (https://github.com/open-mmlab/mmdetection) and MMSegmentation (https://github.com/open-mmlab/mmsegmentation) codebases are released under the Apache-2.0 license.    Openseg.pytorch (https://github.com/openseg-group/openseg.pytorch) codebase is released under the MIT license.

**Limitations.** The presented affinity propagation method is performed under the guidance of the similarities of image intensity and color. Our proposed method may have difficulties in accurately capturing the pairwise affinities under the challenging scenarios like motion blur, occlusions, and cluttered scenes, *etc*. Actually, this is a common problem for many segmentation methods. In the future work, we will explore how to integrate our method into the large-scale foundation models, such as SAM [13], to take advantage of their strong features for more promising segmentation results.

**Broader Impact.** This work presents an effective component for weakly-supervised segmentation with label-efficient annotations. We have demonstrated its effectiveness over three typical label-efficient segmentation tasks. On the positive side, our approach has the potential to benefit a wide variety of real-world applications, such as autonomous vehicles, medical imaging, remote sensing and image editing, which can significantly reduce the labeling costs. On the other side, erroneous predictions in real-world applications (*i.e.*, medical imaging analysis and tasks involving autonomous vehicles) raise the safety issues of human beings. In order to avoid the potentially negative effects, we suggest to adopt a highly stringent security protocol in case that our approach fails to function properly in real-world applications.

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

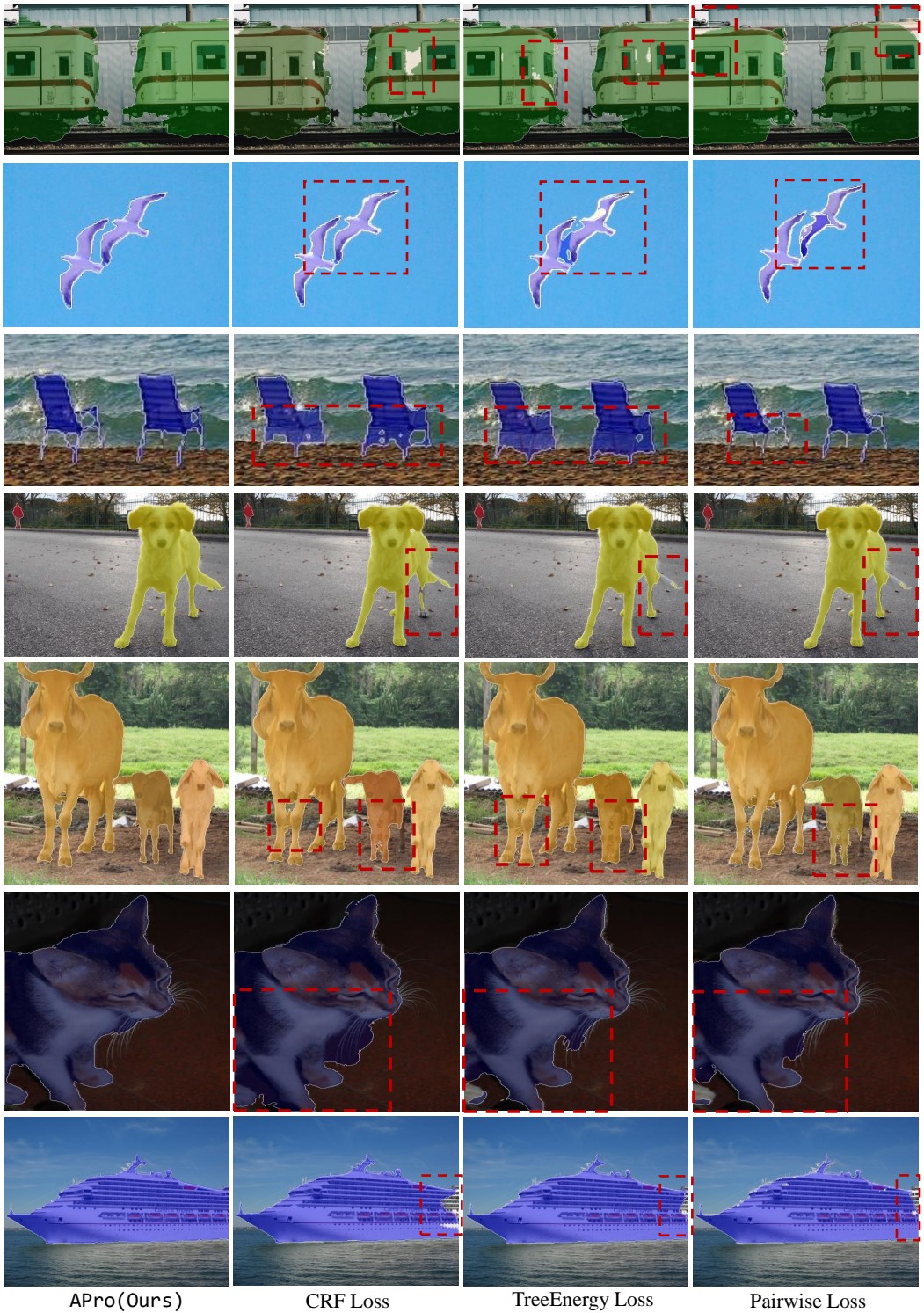

APro(Ours)        CRF Loss        TreeEnergy Loss        Pairwise Loss

Figure A2: Qualitative comparisons on Pascal VOC [9]. We compare our APro approach with CRF loss [14], TreeEnergy loss [7] and Pairwise loss [4] under the SOLOv2 [15] framework. Our method obtains more fine-grained predictions with detail preserved.

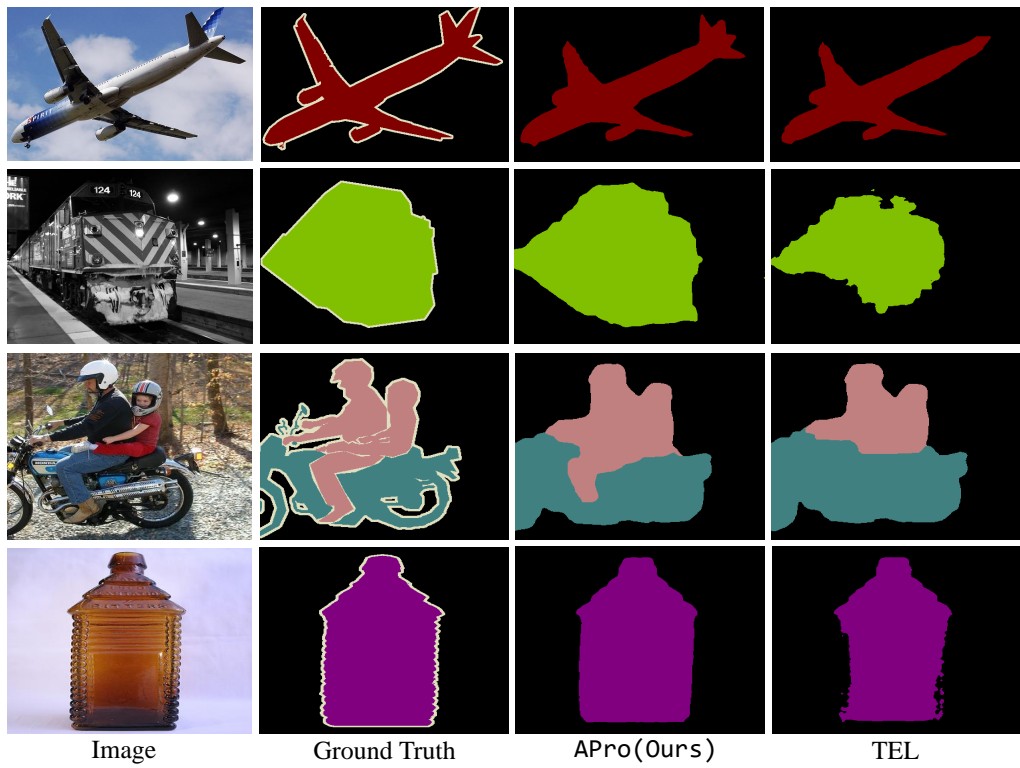

|   |   |   |   |
|---|---|---|---|
| Image | Ground Truth | APro(Ours) | TEL |

Figure A3: Qualitative comparisons on point-supervised semantic segmentation. Compared with the state-of-the-art TEL [7], our method segments objects with more accurate boundaries.

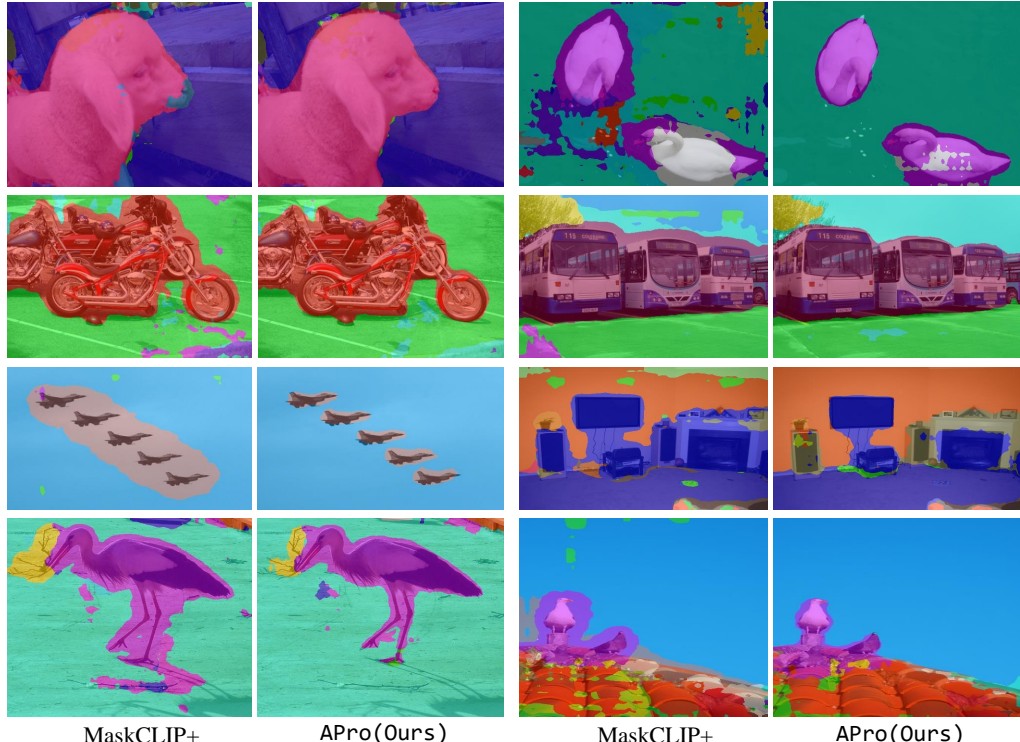

|   |   |   |   |
|---|---|---|---|
| MaskCLIP+ | APro(Ours) | MaskCLIP+ | APro(Ours) |

Figure A4: Visual comparison results on Pascal Context with ViT-B/16 image encoder. Compared with the prior art MaskCLIP+ [1], our method obtains more accurate predictions with fitting boundaries.

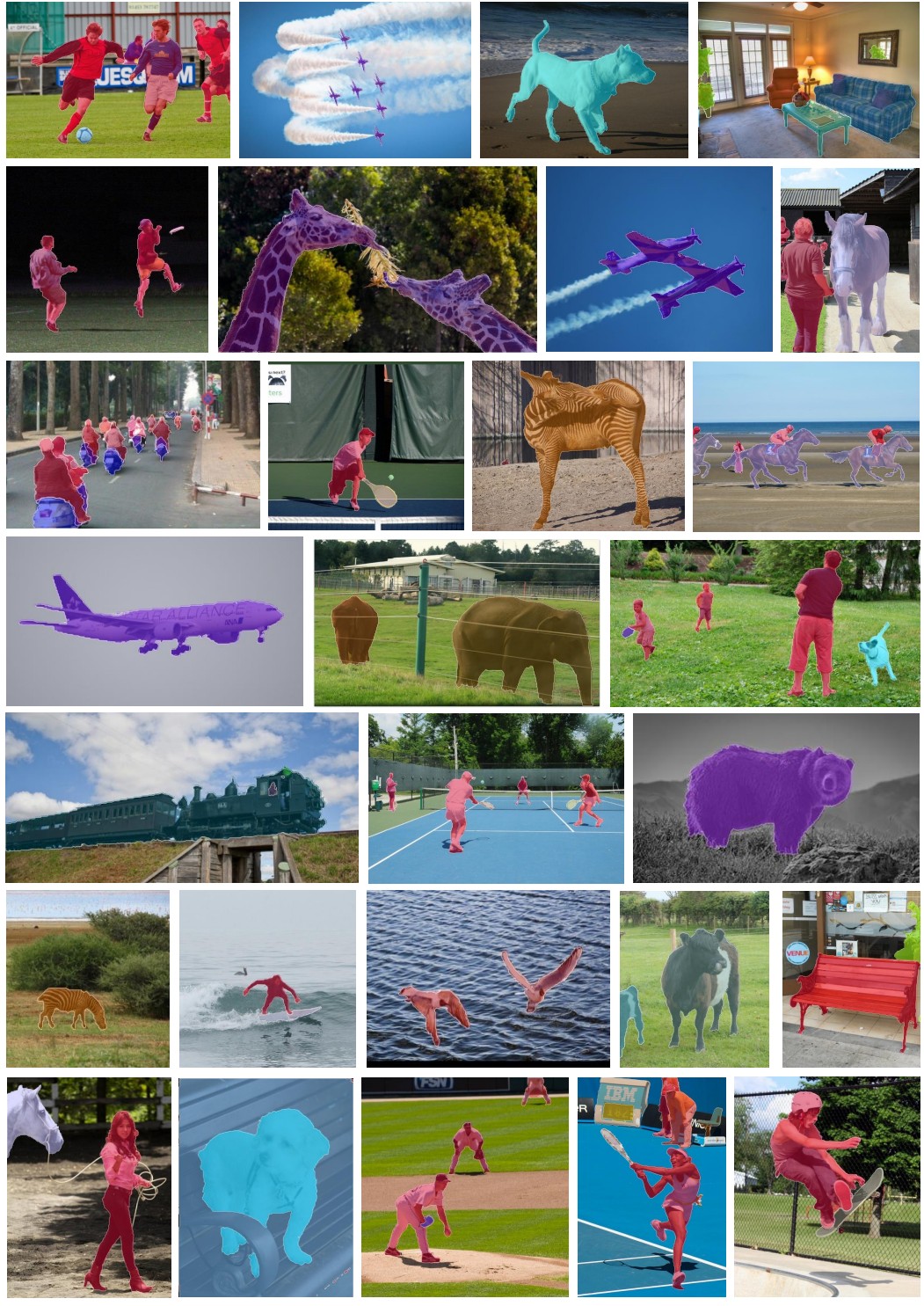

Figure A5: Qualitative results of our `APro` on COCO with ResNet-101 under the SOLOv2 framework upon box-supervised instance segmentation.