# OpenReview forum: "Label-efficient Segmentation via Affinity Propagation"
_NeurIPS.cc/2023/Conference — NeurIPS 2023 poster_

### Official Review · Reviewer_bNsh · 2023-07-04

**Soundness:** 3 good
**Presentation:** 2 fair
**Contribution:** 3 good
**Rating:** 5
**Confidence:** 3

**Summary:**

This paper proposes a novel universal component for weakly-supervised segmentation by formulating it as an affinity propagation process. It simultaneously utilizes a global and a local pairwise affinity term to generate soft pseudo labels. An efficient algorithm is also developed to reduce the computational cost. Experiments on three label-efficient segmentation tasks demonstrate the effectiveness of the proposed method.

**Strengths:**

1. The proposed framework uses both global and local pairwise affinity term and achieves superior performance.
2. The efficient implementation of global affinity propagation can greatly reduce computational cost.
3. The proposed approach can be conveniently plugged into existing segmentation networks.
4. The experiments are abundant, covering many label-efficient segmentation tasks.

**Weaknesses:**

1. This approach seems parameter-sensitive. The slight variation of $\zeta_s$, $\zeta_g$ may lead to notable fluctuation of segmentation performance. Is there consistency in the parameters used across different tasks and datasets?
2. The efficiency of the whole framework is not intuitive. By introducing both global and local pairwise affinity term, is there a significant decrease in efficiency? It would be helpful if authors could present the change of training time with/without APro.


**Questions:**

1. How to determine the number of iterations?
2. The runtime details in Table 7 are not clear, such as iterations, batch size, device...
3. In MaskCLIP, key smoothing and prompt denoising are proposed to refine the pseudo masks. The key smoothing also aims to realize global affinity propagation based on the similarity of key features (not used in the MaskCLIP+ setting). It would be helpful to explore their effects.


**Limitations:**

Please refer to the Weaknesses part.

---

> ### Author Rebuttal · Authors · 2023-08-09
>
> Thank you so much for the careful and thoughtful reviews. Please find what below our itemized responses.
>
> **Q1.  Whether parameters ${\zeta_s}$, ${\zeta_g}$ are consistent across different tasks and datasets.**
>
> R: The parameters ${\zeta_s}$ and ${\zeta_g}$ control the sensitivity to variations in pixel values, and they will impact the segmentation performance. While meticulous tuning of these parameters on different datasets and tasks could lead to improved results, we utilize the same values of them across all tasks and datasets: ${\zeta_s}$ = 0.15, ${\zeta_g}$ = 0.07. The details are shown in Tables A1 and A2 in the Supplementary Material.
>
> **Q2. By introducing both global and local pairwise affinity terms, is there a significant decrease in efficiency?  Presenting the change of training time with/without APro.**
>
> R:  We report the detailed training time below. We conduct the ablation studies on the box-supervised instance segmentation on Pascal VOC.
> |$\quad$Method | Training Time  | &nbsp;&nbsp;AP |
> |:----: | :----: | :----: |
> |Baseline | 3h | 25.9  |
> | +LP | 3.5h | 36.0 |
> | +GP | 4.5h | 37.0 |
> | APro (LP+GP) | 5.2h | 38.1 |
>
> The baseline model without our APro method needs 3h to train. When adding the local operation LP and global operation GP individually,  0.5h and 1.5h additional training time are needed, respectively. When adding both of them, it costs 2.2h additional training time with our efficient implementation. However, without our efficient implementation, the training time would be unbearable.
>
> We will make it clearer in our revision.
>
> **Q3. How to determine the number of iterations?**
>
> R: We perform ablation studies to determine the number of iterations. Our goal is to implement the formulated affinity propagation process efficiently with fewer iterations. The detailed ablation experiments on the impact of varying iteration is provided in Table 6 of our main paper.
>
> **Q4. The runtime details in Table 7 are not clear, such as iterations, batch size, device...**
>
> R: Thanks for your careful comments. The batch size is set to 1, and the experiment is conducted on a single GeForce RTX 3090. The reported runtime represents the average time for one GP process testing duration of an epoch on the Pascal VOC dataset. We perform the runtime comparison under the same settings. We will make it clearer in the revision.
>
> **Q5. Explore the effects of Key Smoothing and Prompt Denosing in MaskCLIP.**
>
> R: Yes, the  Key Smoothing (KS) also aims to realize the global affinity propagation. To better explore their efforts,  we conduct detailed comparisons between the KS and our APro method based on MaskCLIP. The experimental results are shown in the Table below.
>
> |  $\qquad$Method   |  CLIP Model  | Context | COCO  |
> | :------------: | :----------: | :-----: | :---: |
> |    MaskCLIP    |  ResNet-50   |  18.46  | 10.17 |
> |      +KS       |  ResNet-50   |  21.0   | 12.42 |
> | **+APro(Ours)**   |  ResNet-50   |  **21.67**  | **12.70**  |
> |    MaskCLIP    | ResNet-50x16 |  21.57  | 13.55 |
> |      +KS       | ResNet-50x16 |  22.65  | 15.50  |
> |  **+APro(Ours)**   | ResNet-50x16 |  **24.03**  | **16.30**  |
> |    MaskCLIP    |    ViT16     |  21.68  | 12.51 |
> |      +KS       |    ViT16     |  23.87  | 13.79 |
> |     +KS+PD     |    ViT16     |  25.45  | 14.62 |
> |  **+APro(Ours)**   |    ViT16     |  **28.91**  | **16.69** |
> | **+APro(Ours)+PD** |    ViT16     |  **29.42**  | **16.71** |
>
> Both KS and our APro method bring performance gains. Compared with KS, our APro achieves better performance with different CLIP-based models. Especially, for ViT16-based model, our approach outperforms KS by +5.04\% mIoU on Pascal Context and +2.90\% mIoU on COCO, repectively. Equipped with Prompt Denoising (PD),  the models could achieve further improvements.
>
> We have the following further discussions:
> Key Smoothing relies on the calculation of key feature similarities, which predominantly stems from high-level features of CLIP and computes pairwise terms within each pair of patches. Compared with KeySmoothing of MaskCLIP,  our method is built on a tree-based graph derived from lower-level images, which is capable of reflecting finer topological details. Furthermore, we design an efficient implementation that eliminates the need to compute similarities individually, significantly reducing time complexity.
>
> We will add the above discussions in our revision, and strengthen our introduction and experimental sections.

---

> > ### Comment · Reviewer_bNsh · 2023-08-12
> >
> > Thanks for your rebuttal. The rebuttal has addressed most of my concerns and the discussion is instructive. I will keep my original rating.

---

> > > ### Author Response · Authors · 2023-08-12
> > > **Thanks for your response**
> > >
> > > Dear Reviewer bNsh,
> > >
> > > Thanks very much for your confirmation and your time.

---

### Official Review · Reviewer_8pug · 2023-07-04

**Soundness:** 3 good
**Presentation:** 3 good
**Contribution:** 3 good
**Rating:** 5
**Confidence:** 3

**Summary:**

This paper proposes an affinity propagation algorithm for weakly supervised segmentation. Given image segmentation data with only sparse box/point/scribbles annotations, the proposed method propagates these annotations to other pixels as pseudo masks for training. Global affinity and local affinity are proposed to capture global pairwise potential and local connectivity respectively. The authors propose an efficient implementation for the propagation of these two types of affinity which makes training practical. The final model performs better than prior works in PASCAL VOC and COCO with partial annotations.

**Strengths:**

- The proposed framework is simple and effective. Both global and local affinity propagation are intuitive, and the resultant method achieves strong performance compared to existing works without bells and whistles. This shows the effectiveness of the proposed affinity propagation mechanism.

- The authors propose an efficient implementation for the affinity propagation algorithm which is crucial for making training with this method practical. The authors claim that it is five times faster than mean-field [29]. This fast algorithm is also potentially useful for future work in other directions.


**Weaknesses:**

- The affinities are defined on pixel intensity which can be limiting as it is sensitive to lighting and low-level noises. It might help to incorporate deep features from self-supervised, pre-trained networks (e.g., MAE).

- Another direction for segmentation is open-vocabulary segmentation which can benefit from scaling and data engines (e.g., SAM). The proposed method does not seem to be scalable as the affinity is fixed to low-level color differences.

- Missing related work and discussion: Learning Pixel-level Semantic Affinity with Image-level Supervision for Weakly Supervised Semantic Segmentation, CVPR 2018


**Questions:**

- Will the code be released? I see no mention of this in the paper. It would help the community if an open-source version is available, especially for the non-trivial GPU implementation of affinity propagation.

- When comparing speed with mean-field [29], are both algorithms running on GPU?


**Limitations:**

Limitations are mentioned in Section D in the supplementary material. The authors mention that the use of only image intensity to compute affinity is a limitation which I agree with.

---

> ### Author Rebuttal · Authors · 2023-08-09
>
> Thank you so much for the constructive and insightful comments.  Please find what below our itemized responses.
>
> **Q1. About incorporating deep features from self-supervised, pre-trained networks (e.g. MAE).**
>
> R: We have tested the incorporation with the deep features of the network during training. It is unstable to achieve reasonable performance in our experiments. In the future work,  we will consider to incorporate deep features from the pre-trained MAE, DINO/DINO-V2  models, as you mentioned. These pre-trained models have exhibited strong generalization capability, and we believe they could make our method more powerful.
>
> **Q2. For open-vocabulary segmentation, the proposed method does not seem to be scalable as the affinity is fixed to low-level color differences.**
>
> R: Yes, our approach may not perform well in the task of open-vocabulary segmentation. As you suggested,  we could incorporate the pre-trained models such as DINO/DINOv2, MAE, CLIP or Stabel diffusion to improve the scalability of our method.
>
> In addition, our method can be extended to multimodality-based affinities, i.e., between vision and language, to achieve more accurate vision-language alignment for VL tasks. This is an interesting research direction for our further work.
>
> **Q3. Missing related  work and discussion of the CVPR2018 work.**
>
> R: The work [1] (CVPR2018) also involves affinity modeling operations, while the formulations and pipelines of them are different from our approach. Especially, the work [1] adopts the sparse random walk for long-range operations, which relies on the affinity transition probability. Different from it, our method constructs the Minimum Spanning Tree (MST) on the whole image and performs the formulated affinity propagation process. To reduce the computation cost, we deliberately devise a Lazy Propagation scheme for fast implementation. Please kindly refer to our **responses to Reviewer bFj5** for more comprehensive analyses and comparisons with the other existing methods.
>
> [1]  Learning Pixel-level Semantic Affinity with Image-level Supervision for Weakly Supervised Semantic Segmentation, CVPR 2018.
>
> **Q4. Will the code be released?**
>
> R: For sure, we will release our full source code to make contributions to the community, including the GPU implementation of affinity propagation and complete codes on each task, so that peers can easily reproduce our results.
>
> **Q5. When comparing speed with mean-filed, are both algorithms running on GPU.**
>
> R: Yes, both methods are performed on the GPU device. The original mean-field [29] is based on the pixels of the whole image, and we compare the GPU version [4] of mean-filed with local kernel under the same settings. We will make it clearer in the revision.

---

> > ### Comment · Reviewer_8pug · 2023-08-10
> >
> > Thank you for the response. It addressed my concerns and I have no further questions.

---

> > > ### Author Response · Authors · 2023-08-11
> > > **Thanks for your response**
> > >
> > > Dear Reviewer 8pug,
> > >
> > > Thanks very much for your confirmation and your time!

---

### Official Review · Reviewer_acqT · 2023-07-04

**Soundness:** 3 good
**Presentation:** 2 fair
**Contribution:** 3 good
**Rating:** 5
**Confidence:** 4

**Summary:**

This paper develops a weakly-supervised segmentation framework based on affinity propagation. It overcomes the drawback of simply modeling neighboring pairwise potentials and proposes both global and local affinity terms to generate pseudo labels. The authors demonstrate the effectiveness of the proposed method in box-supervised instance segmentation, point/scribble-supervised semantic segmentation, and CLIP-guided semantic segmentation tasks.

**Strengths:**

Originality: The paper proposes two kinds of pairwise affinity propagation which are novel for weakly-supervised image segmentation settings. The global affinity propagation employs a minimum spanning tree to remove the edge with a large distance to obtain the tree-based sparse graph, which explicitly captures long-range dependency, while local affinity propagation leverages the conventional Gaussian kernel.  The combination of the two enhances the performances based on Table 4.

Quality: The paper itself is self-contained and includes sufficient experimental setup and details.

Clarity: The paper is well-written but it might be hard to follow. It would be hard to understand the context of the proposed method for my initial reading and still creates some confusion after repeated reading. Figures are suggested to enhance the clarity of certain concepts developed in this paper.

Significance: I give high significance to this paper, as it proposes an effective and convincing method for a well-known and challenging problem.


**Weaknesses:**

Personally, the biggest concern of this paper is the clarification. Basically, the authors devised two affinity scheme to create long-range and short-distance dependencies and both of them uses the same framework as described in Equ. 2. However, it would be very hard to understand the details of the global one. Authors are suggested to make a graphical illustration to demonstrate the idea.  Also, it still remains confusion about the method, as will be asked in the next section.


**Questions:**

It would be appreciated if the authors can kindly reply to my questions and concerns. I will consider raising my evaluation.

- The whole framework of affination is defined on the classic CRF model, as cited in line 110. However, it is noted that in [33], the energy function is the sum of unaries and the sum of pairwise potentials. The CRF model in Equ. 2 turns out to be the sum of products between unary and pairwise terms. If understand correctly, this formulation can be interpreted as the weighted sum of unaries defined on neighbor pixels, with weights calculated from pairwise terms. I am wondering how the authors draw an equivalence between the two formulations.

- What is the $\mathcal{L}_{g}$ used in Equ. 1? is it a partial cross-entropy? how is it defined with non-annotated pixels? The unary used in Equ. 2 is then the output distribution of only labeled pixels or all pixels?

- In 3.2.1, the node in line 127 is defined as a pixel right? why the set of edges can have N-1 elements, as defined in line 132

- How do authors set the degree of similarity defined on line 143? is it unique for all nodes?

- Can the authors explain why the tree-based model preserves topology?

- For the local affinity propagation, I always consider Equ. 6 as a sum of neighbor pixels weighted by the intensities, similar to the conventional CRF model [33]. However, in [33], a spatial smoothness term is added. The authors want to explain how their method preserves spatial smoothness.

- Based on Equ.6  I don't see any necessity of iterating it (there are even no iterative parameters inside). What does it mean by saying iterating local affinity leads to better performance?

- what is the transmission cost?


- In 3.2.1 and 3.2.1, the authors then obtain two sets of pseudo labels, $y^{g}$ and $y^{s}$. How do they exactly train the network with these two sets of labels? How to merge them when using them together?




**Limitations:**

The authors adequately addressed the limitations.

---

> ### Author Rebuttal · Authors · 2023-08-09
>
> Thank you so much for acknowledging the strength of our method. We have tried our best to clarify each issue.  Please find what below our itemized responses.
>
> **Q1. The details of the global one and making a graphical illustration.**
>
> R: To facilitate comprehension, we provide a detailed graphical illustration **(Figure 2 in the PDF)** to describe our global affinity propagation process.
>
> Initially, an input image is represented as a 4-connected planar graph. Subsequently, the Minimum Spanning Tree (MST) is constructed based on the edge weights to obtain the tree-based graph $\mathcal{G}\_T$. $\psi_g(x_i,x_j)$ is calculated as $exp(-d)$,  where $d$ is the maximum value along the path $E_{i,j}$ from node $x_i$ to node $x_j$. This pairwise similarity $\psi_g(x_i,x_j)$  is then multiplied by the unary term $\phi(x_j)$ to obtain soft pseudo predictions  $y_i^g$.
>
> Note that Figure 2 serves purely as a visual illustration to understand our method. In the implementation, it is unnecessary to compute $\psi_g$ explicitly. As detailed in Section 3.3, we alternatively design a lazy propagation scheme to efficiently update these values.
>
> **Q2.  About the formulations of Equ.2 and the classic CRF model.**
>
> R: Yes, you are right. Our definition of the affinity propagation process does not strictly adhere to the classic CRF model[29]. The energy function in [29] consists of the sum of unary terms and pairwise potential; however, the problem solving process is rather complex. As our aim is to obtain a refined pseudo label, we adopt the general concept of unary and pairwise terms, with the intention of integrating their benefits in a more accessible and straightforward manner.
>
> **Q3.  About the  $\mathcal{L}_g$ used in Equ.1 and the unary used in Equ.2.**
>
> R: The definition of $\mathcal{L}_g$ depends on the form of supervision. For point or scribble forms, the sparsely labeled region lies in the object, which is suitable for partial cross-entropy loss. It has no impact on non-annotated pixels. While, for labeled bounding box supervision, it is uncertain whether the pixels within the box truly belong to the object. So we adopt box projection loss, which constraints the predictions within the labeled box. In our paper, we listed the loss function for each unary term in the section of Implementation Details (lines 210, 245-248) for different weakly-supervised tasks.
>
> Moreover, the unary term used in Equ. 2 is the network prediction of all pixels.
>
> We will make our revision clearer.
>
> **Q4. About node in line 127 and N-1 edges.**
>
> R: Yes, the node corresponds to a pixel. The constructed MST is an acyclic subgraph of the original 4-connected planar graph that includes all vertices.  A spanning tree connects all nodes of an image, and there is a unique and simple path between any two nodes. Therefore, the MST-based graph with N vertices requires N-1 edges to maintain connectivity without cycles. **Figure 2 in the PDF** also gives a simple illustration of a constructed MST.
>
> **Q5. About the degree of similarity defined in line 143.**
>
> R: The degree of similarity $\zeta_g$ is a hyper-parameter in our approach. We conduct ablation study on it in Table A1 of the supplementary material and select $\zeta_g$ = 0.07, which remains constant for all nodes.
>
> **Q6. Explain why the tree-based model preserves topology.**
>
> R: Our approach initially represents an image as a 4-connected planar graph with pixel similarity measured via the edge weight of adjacent nodes. The MST is constructed by edge pruning, which preferentially preserves edges of smaller weight, i.e., adjacent vertices with higher pixel similarity.
>
> Similar pixels are usually located inside or on the surface of an object, whereas larger pixel differences appear across distinct objects. In other words, there are more edges within an object and fewer edges connecting different objects.  Thus, the MST can capture an image's topological structure. For better comprehension, we provide two visual examples in **Figure 4 of the PDF**.
>
> **Q7.  How the local affinity propagation preserves spatial smoothness?**
>
> R: In the conventional CRF model, the pairwise potentials involve relationships between each pixel and all the other pixels, necessitating the inclusion of a spatial smoothness term to maintain spatial coherence. While our local affinity propagation method applies pairwise affinity within a local region surrounding each pixel, such as a 3x3 or 5x5 kernel. This consideration of local domain pixels implicitly signifies spatial smoothness, indicating that nearby pixels likely belong to the same class. The detailed experimental comparison is provided below. Incorporating Spatial Position into our LP yields no performance improvement while bringing an additional hyperparameter.
>
> |$\qquad$Method|AP|
> |:-:|:-:|
> |LP|36.0|
> |LP + Spatial Position|34.8|
>
> **Q8. About the iteration of the local affinity.**
>
> R: Thanks for pointing out this issue. Upon obtaining the refined pseudo label $y_i^s$, we treat it as a new unary term $\phi(x)$ to iterate it. Table 6 in the main paper demonstrates the effectiveness of the iterating process.
>
> **Q9. What is the transmission cost?**
>
> R: In Section 3.3, we define the maximum $w$ of the path through any two nodes in the tree-based graph as the transmission cost $C$.  For example, in the green dashed box of **Figure 2 in the PDF**, the transmission cost of $x_0$ and $x_3$ is 3*${\zeta_g}^2$. Section A.1 in our Supplementary Material also provides detailed proof of transmission cost.
>
> **Q10. How to train the network with labels $y^g$ and $y^s$?**
>
> R:  We assign each $y_i$ from GP and LP to the network prediction $p_i$, and employ the distance measurement function as the objective for unlabeled regions $\Omega _u$. Simple L1 distance is empirically adopted in our implementation.
>
> We described it in Section 3.1 (line122-line124) of the main paper. We will make it clearer in the revision.

---

### Official Review · Reviewer_4J6P · 2023-07-13

**Soundness:** 3 good
**Presentation:** 3 good
**Contribution:** 2 fair
**Rating:** 5
**Confidence:** 4

**Summary:**

This paper proposes an affinity propagation method within local and global perspectives to improve the pseudo labels generated by the model for the parts without GT masks. Meanwhile, they then propose an efficient implementation to solve the heavy computation by graph modeling.  The authors conduct experiments on several benchmarks to showcase their improvements on point/scribble/bbox-level weakly-supervised semantic segmentation, acting as a plug-in module to enhance their segmentation performance. And competitive performances are obtained naturally.

**Strengths:**

1. The writing of this paper is easy to follow.
2. Extensive experiments are studied and analyzed to present the proposed method.
3. The final performances are competitive.

**Weaknesses:**

1. To my concern, affinity propagation exploration has been studied widely in the weakly-supervised learning community, such as [1]. The overall novelty of this paper sounds limited because their method has a lot of overlappings with previous method, though the author adapt a graph modeling to analyze and model the propagation procedure. It is important to demonstrate the main differences from the proposed method with others.

    [1] Learning Affinity from Attention: End-to-End Weakly-Supervised Semantic Segmentation with Transformers, CVPR2022

2. From the eq3 and eq5, it is hard to get the point, of how they define the global and local respectively since the main differences lie in that a max operation is performed in eq3 and the points-set in eq5 is local but not detailed clearly.

3. The motivation for adapting Gaussian to model the local part in this paper should present more reasonably, for example, by giving some experiments directly to see that the Gaussian methods indeed capture the local receptive features.

4. Despite the point/scribble/bbox-level weakly supervised semantic segmentation, how about the results when performing on semi-/image-level supervised semantic segmentation tasks to validate that the proposed method indeed helps to improve the soft-pseudo label for the unlabeled pixels?

5. Some important references are missing below while these methods may discuss in the paper too.

    [2] Reducing Information Bottleneck for Weakly Supervised Semantic Segmentation, NeurIPS 2021

    [3] Expansion and Shrinkage of Localization for Weakly-Supervised Semantic Segmentation, NeurIPS 2022

    [4] Revisiting Weak-to-Strong Consistency in Semi-Supervised Semantic Segmentation, CVPR2023

    [5] Semi-Supervised Semantic Segmentation With Error Localization Network, CVPR2022

**Questions:**

Please refer to the weakness part.

**Limitations:**

Despite the limited novelty in this phase, this paper can be stronger when the authors well address my questions above.

---

> ### Author Rebuttal · Authors · 2023-08-09
>
> Thank you so much for the thoughtful comments.  Please find what below our itemized responses.
>
> **Q1. Main differences with the existing approaches.**
>
> R: Different from the existing approaches, our method considers object topology and captures fine-grained global affinity through an efficient implementation. Unfortunately, the existing methods fall short in achieving such an objective.
>
> Besides, our method is a general plug-in module for various label-efficient segmentation tasks, which does not require any modification on the network itself. Extensive experiments demonstrate the effectiveness of our approach in generating high-quality pseudo mask predictions across different tasks and datasets.
>
> Please kindly refer to **our responses to reviewer bFj5** for more comprehensive analyses on the existing methods, including AFA [1].
>
> [1] Learning Affinity from Attention: End-to-End Weakly-Supervised Semantic Segmentation with Transformers, CVPR2022.
>
> **Q2. About the differences between the global and local  in Eq.3 and Eq.5.**
>
> R: The formulations of global (Eq.3) and local (Eq.5)  have the similar form, while they are defined in different scopes. In detail, their receptive fields for affinity propagation are totally different.
>
> The global pairwise term contains all the nodes $ \mathcal{V}$ through the tree-based graph $\mathcal{G}_T$, which are implemented by the Minimum Spanning Tree. In contrast, the local pairwise term is limited to a local area, such as 3x3 or 5x5, which is implemented by the Gaussian kernel.
>
> To better illustrate the detailed process of the global and local operations, we provide **Figure 1 in the PDF**. For visual comparisons of pairwise affinity maps, Figure 3 in the main paper also shows the global and local affinity maps, denoted as "GP" and "LP", respectively.
>
> In calculating the global pairwise term, we present the distance-insensitive max affinity function to ensure that the similarity does not diminish abruptly with the increase of distance along the path of the spanning tree (see **Figure 2 in the PDF** for the details).
>
> **Q3. The motivation for adapting Gaussian to model the local part should present more reasonably. Giving some experiments directly to see.**
>
> R:  Spatially adjacent pixels are more likely to share the same label, which inspired us to define the local Gaussian kernel within a fixed window, such as 3x3 or 5x5. Such a neighboring receptive field allows the focus to be placed on local texture and shape features rather than global ones. Intuitively, we present the visualizations in **Figure 3 in the PDF** to demonstrate the effectiveness of this local pairwise term.
>
> One can see that the predictions become smoother after local affinity propagation (LP), indicating an enhancement in local consistency by capturing the local receptive characteristics.
>
> **Q4. How about the performance on semi-/image-level supervised semantic segmentation?**
>
> R: Thanks for your suggestions. We have conducted the image-level semantic segmentation task on Pascal VOC2012 dataset based on the framework of AFA [1]. The comparison results are shown below. Our method can further obtain +1.5\% mIoU gain over AFA[1].
> |&nbsp;&nbsp;&nbsp;Method | dataset | $\quad$&nbsp;&nbsp;&nbsp; mIoU |
> |:-:| :-: |:-:|
> |  AFA[1] | VOC2012 | 62.6 |
> | +APro(Ours) | VOC2012 | $\qquad$64.1($\uparrow$1.5) |
>
> We will perform more experiments on recent image-level methods, such as ESOL(NIPS2022) and ToCo(CVPR2023),  as well as semi-supervised approaches [4][5], and add the results into the revised manuscript to further show the effectiveness of our method.
>
> **Q5. Some important references are missing below while these methods may discuss in the paper too.**
>
> R: Thanks for your suggestions.  Methods [2] and [3] are about image-level supervised semantic segmentation methods, while [4] and [5] are related to semi-supervised semantic segmentation. Specifically, RIB [2] adopts the information bottleneck principle to interpret the partial localization issue in trained classifier. ESOL[3] proposed a new training pipeline,  with a "Divide-and-Conquer" manner to address the partial localization issue of the CAM method by introducing a deformable transformation operation. It is worth noting that we have already cited [3] in our manuscript. On the other hand, ELN [5] aims to deal with errors on pseudo labels and UniMatch [4] presents the weak-to-strong consistency regularization framework from FixMatch.
>
> We will cite and discuss these works in our revised manuscript.
>
> &emsp;
>
> [2] Reducing Information Bottleneck for Weakly Supervised Semantic Segmentation, NeurIPS 2021.
>
> [3] Expansion and Shrinkage of Localization for Weakly-Supervised Semantic Segmentation, NeurIPS 2022.
>
> [4] Revisiting Weak-to-Strong Consistency in Semi-Supervised Semantic Segmentation, CVPR2023.
>
> [5] Semi-Supervised Semantic Segmentation With Error Localization Network, CVPR2022.

---

> > ### Comment · Reviewer_4J6P · 2023-08-18
> > **Rebuttal Response**
> >
> > I would like to thank the authors' detailed response and more explanation. To my perspective, most of my concerns are well addressed. And I would like to suggest that the authors conduct more convincing results on other segmentation-related tasks to further enhance their plug-in role. Overall, I will raise up my initial score to 'borderline accept' and suggest acceptance.

---

> > > ### Author Response · Authors · 2023-08-18
> > > **Thanks for your response**
> > >
> > > Dear Reviewer 4J6P,
> > >
> > > Thanks very much for your positive feedback and further suggestions!  We'll include more convincing results in our revision as you suggested.

---

### Official Review · Reviewer_bFj5 · 2023-07-22

**Soundness:** 3 good
**Presentation:** 3 good
**Contribution:** 2 fair
**Rating:** 4
**Confidence:** 4

**Summary:**

This paper utilizes local and global pairwise affinity terms to generate accurate soft pseudo labels and incorporates an efficient algorithm to reduce computational costs. Experimental results demonstrate the approach's superior performance in various segmentation tasks.

**Strengths:**

Experimental results demonstrate the approach's superior performance in various segmentation tasks.
The paper is easy to understand.

**Weaknesses:**

The affinity methods are commonly used in weakly supervised segmentation tasks, such as 'Learning Affinity from Attention: End-to-End Weakly-Supervised SemanticSegmentation with Transformers
Affinity Attention Graph Neural Network for Weakly Supervised Semantic Segmentation.
Learning Pixel-level Semantic Affinity with Image-level Supervision for Weakly Supervised Semantic Segmentation
Weakly Supervised Learning of Instance Segmentation with Inter-pixel Relations.'

The novelty of their method might be limited compared to existing affinity-based approaches. To stand out, the author should clearly highlight the differences between their method and the mentioned ones. It could be in terms of the formulation of the affinity modeling task, the incorporation of local and global pairwise affinity terms, the generation of accurate soft pseudo labels, or the development of an efficient algorithm to reduce computational costs.

Providing a thorough comparison with these existing methods would help readers understand the unique contributions of the proposed approach and its advantages over previous approaches. By highlighting these differences, the paper can demonstrate why their method is valuable and relevant in the context of weakly supervised segmentation tasks.

**Questions:**

To make their contribution more apparent, the author should provide a comprehensive comparison of the proposed method's performance with previous approaches of the generated pseudo ground truth. This comparison would highlight the advantages and improvements of their approach over existing methods in terms of accuracy, efficiency, and other relevant metrics.

By including a thorough analysis of the pseudo ground truth performance, readers can better understand the strengths of the proposed approach and how it outperforms or complements existing techniques.

**Limitations:**

See Weaknesses

---

> ### Author Rebuttal · Authors · 2023-08-09
>
> Thank you so much for the constructive comments. Please find what below our itemized responses.
>
> **Q1. The comprehensive analysis and comparisons with the existing affinity-based methods.**
>
> R:  Previous works [1-4] involve affinity modeling, and our proposed approach is largely different from them in formulation and pipeline.  Specifically,  [1] adopts the random walk and local pixel-adaptive refinement (PAR) to propagate the affinity. [2] introduces the Affinity CNN to convert images into graphs, using a GNN-based affinity attention module for affinity modeling. [3,4] leverage the random walk to capture long-range affinity with local operations. While they struggle with fine-grained global semantic affinity, hindering the generation of accurate pseudo-mask labels. As noted in [3], detailed global affinity modeling requires substantial computational costs.
>
> In contrast, we present the unified affinity propagation formulation for both global and local affinity modeling. The global affinity utilizes the acyclic Minimum Spanning Tree to perform fine-grained affinity propagation, which considers all nodes for a comprehensive context understanding with object topology. On the other hand, the local term employs the kernel-based propagation in a complementary role, which focuses on the nearby area to attain spatial smoothness. To efficiently perform this procedure, we deliberately design a Lazy Propagation scheme for fast implementation, which avoids excessive computations.
>
> Furthermore, we devise a general plug-in module for various label-efficient segmentation tasks, which does not require any modification of the network itself. In comparison, the previous works[1][2][3][4] are mainly designed for single weakly-supervised segmentation tasks with some customized designs.
>
> To demonstrate the effectiveness of our method, we further conduct detailed performance comparisons with the recent works [1][2].
> Firstly, we implement our method based on the AFA[1] framework for image-level supervised semantic segmentation on Pascal VOC 2012. The experimental settings are the same as [1] for fair comparison. The results are reported below.
> |&nbsp;&nbsp;&nbsp;Methods | dataset | $\quad$&nbsp;&nbsp;&nbsp; mIoU |
> |:-:| :-: |:-:|
> |  AFA[1] | VOC2012 | 62.6 |
> | +APro(Ours) | VOC2012 | $\qquad$64.1($\uparrow$1.5) |
>
>
> We notice that the local pixel-adaptive refinement (PAR) in AFA [1] is similar to our proposed local affinity term (LP). Both of them are kernel-based methods based on the input image, however, their formulations and implementations are different. In particular, we compare our method with PAR [1] in weakly box-supervised instance segmentation settings on VOC2012 and COCO.
> |  $\qquad$ Methods | &nbsp;&nbsp;dataset | $\quad$&nbsp;&nbsp;AP| $\quad$AP_50 | $\quad$AP_75  |
> |:--: | :--: | :--: | :--: | :--: |
> |Baseline +PAR[1]| VOC2012| 34.6 | 63.6| 33.9 |
> |Baseline + LP(Ours)| VOC2012| 36.0 ($\uparrow$1.4) | 64.3($\uparrow$0.7) | 35.6 ($\uparrow$1.7)|
> |Baseline + APro(Ours)| VOC2012| 38.1 ($\uparrow$3.5) | 66.1 ($\uparrow$2.5)| 39.1 ($\uparrow$5.2) |
> |Baseline + PAR[1] | COCO| 30.5 | 53.1 | 30.6 |
> |Baseline + LP(Ours) | COCO|  31.6 ($\uparrow$1.1)  |  53.2 ($\uparrow$0.1)  |   32.2 ($\uparrow$1.6) |
> |Baseline + APro(Ours) | COCO| 33.0 ($\uparrow$2.5)| 55.2 ($\uparrow$2.1) | 33.6 ($\uparrow$3.0)|
>
> Compared with PAR [1], our approach obtains consistent performance gains.
>
> Secondly, we compare our method with A$^2$GNN[2] under its designed point/scribble-supervised semantic segmentation and box-supervised instance segmentation settings.
>
> Methods | backbone |Supervision|Multi-stage | CRF | mIOU |
> |:----: | :----: | :----: | :----: | :----: | :----: |
> |A$^2$GNN[2] | DeeplabV2 | Point| √  | √ | 66.8 |
> |APro(Ours)| Tree-FCN | Point| x  |  x  | 67.7 |
> |A$^2$GNN[2]| Tree-FCN | Scrrible| √ |  √  | 76.2 |
> |APro(Ours)| Tree-FCN | Scrrible| x  |  x  | 76.6 |
>
> Though A$^2$GNN has achieved competitive results, it needs multi-stage training and CRF refinement in the post-processing to yield accurate mask predictions. In contrast, our method is an end-to-end training framework without the CRF post-processing.
>
> Regarding the weakly box-supervised instance segmentation task,  our APro outperforms A$^2$GNN by a large margin, with +15.3\% AP_75 on VOC2012 and +13.4\% AP on the large-scale COCO dataset.
> | Methods | dataset | backbone | AP| AP_50 | AP_75  |
> |:----: | :----: | :----: | :----: | :----: | :----: |
> | A$^2$GNN[2] | VOC2012 | r101  |  -  |  59.1  | 27.4 |
> | APro(Ours) | VOC2012 | r50 |  38.1 | 66.1 | 39.1 |
> | APro(Ours) | VOC2012 |r101 |  **40.6** | **68.5**| **42.7** |
> | A$^2$GNN[2] | COCO |  r101 | 20.9|43.9 |17.8|
> | APro(Ours) | COCO |  r50 |33.0|55.2|33.6|
> | APro(Ours) | COCO |  r101 |**34.3**|**57.0**|**35.3**|
>
> The above results indicate that our method is able to model the fine-grained affinity and obtain accurate mask predictions across different label-efficient segmentation tasks.
>
> We will incorporate the above discussions in our revised manuscript.
>
> &emsp;
>
> [1] Learning Affinity from Attention: End-to-End Weakly-Supervised Semantic Segmentation with Transformers, CVPR2022.
>
> [2] Affinity Attention Graph Neural Network for Weakly Supervised Semantic Segmentation, TPAMI2021.
>
> [3] Learning Pixel-level Semantic Affinity with Image-level Supervision for Weakly Supervised Semantic Segmentation, CVPR2018.
>
> [4] Weakly Supervised Learning of Instance Segmentation with Inter-pixel Relations, CVPR2019.

---

### Author Rebuttal · Authors · 2023-08-09

**General Response**

We express our gratitude to all reviewers for their insightful comments, which significantly strengthen our paper. We will revise our manuscript accordingly.

As three out of the five reviewers concern the differences between our work and some existing works[1-3], we'd like to first clarify that our formulation and process are different from them. The works in [1,3] are based on the random walk and local operation, and [2] utilizes the sparsely weighted graphs with a GNN-based affinity attention module. In contrast to them, our presented method models the semantic affinity through the formulated affinity propagation processes, which takes advantage of both the global affinity model using the Minimum Spanning Tree as well as the complementary local one with Gaussian kernel.

Besides, our work aims to devise a general and efficient plug-in module for various label-efficient segmentation tasks without the need to modify the network itself.  Compared with the existing methods, our approach is able to capture fine-grained affinity to generate accurate mask labels. For a detailed discussion and performance comparisons, please kindly refer to our **Response to Reviewer bFj5**.

To reproduce our results, we will release the full source code of our presented affinity propagation with efficient implementation for each label-efficient task.

In the following, we address the specific concerns point by point. The corresponding figures and captions are included in **the submitted PDF file**. Please feel free to check.

&emsp;

[1] Learning Affinity from Attention: End-to-End Weakly-Supervised Semantic Segmentation with Transformers, CVPR2022.

[2] Affinity Attention Graph Neural Network for Weakly Supervised Semantic Segmentation, TPAMI2021.

[3] Learning Pixel-level Semantic Affinity with Image-level Supervision for Weakly Supervised Semantic Segmentation, CVPR2018.

---

### Decision · Program_Chairs · 2023-09-21

**Decision:**

Accept (poster)

**Comment:**

This paper received four Borderline Accept recommendations and one Borderline Reject (no update after the initial rating).

This work presents a semi-supervised learning approach for image segmentation with efficient label propagation has been proposed.

One main concern is about the significance in novelty as label propagation has been widely adopted in semi-supervised learning. The authors claimed to have presented new algorithms with efficient global and local affinity maps.

From a historical view, the weakly-supervised setting typically refers to the setting where the image-level labels are given without the pixel-level supervision.

Vezhnevets, A., Buhmann, J.M., "Towards weakly supervised semantic segmentation by means of multiple instance and multitask learning", CVPR 2010.
D. Pathak, E. Shelhamer, J. Long, and T. Darrell, “Fully convolutional multi-class multiple instance learning,” arXiv:1412.7144, 2014.
D. Pathak, P. Kr€ahenbuhl, and T. Darrell, “Constrained convolutional neural networks for weakly supervised segmentation,” ICCV, 2015.
P. O. Pinheiro and R. Collobert, “From image-level to pixel-level labeling with convolutional networks,” ICCV 2015.
G. Papandreou, L.-C. Chen, K. Murphy, and A. L. Yuille, “Weakly- and semi-supervised learning of a DCNN for semantic image segmentation,”
ICCV, 2015.

Label propagation has been widely initiated in the semi-supervised learning setting. It is unfortunate that the many of the recent deep learning papers fail to trace back to origins of the literature. For example, the work by Xiaojin Zhu is important but has been ignored:
"Semi-supervised learning literature survey", Xiaojin Zhu.
"Semi-supervised learning using gaussian fields and harmonic functions", X Zhu, Z Ghahramani, JD Lafferty, ICML 2003.

In some of the related works, the term of "weak-supervised" has been extended to include what was traditionally considered as "semi-supervised". This creates confusion in the literature. The authors are recommended to make the definition clear in the revision by better connecting to the literature.